# Dual Critic Reinforcement Learning under Partial Observability

**Jinqiu Li**[1,2]**, Enmin Zhao**[1,2]**, Tong Wei**[3]**, Junliang Xing**[3*]**, Shiming Xiang**[1,2*]
[1]Institute of Automation, Chinese Academy of Sciences
[2]School of Artificial Intelligence, University of Chinese Academy of Sciences
[3]Department of Computer Science and Technology, Tsinghua University
{lijinqiu2021, zhaoenmin2018}@ia.ac.cn, wt22@mails.tsinghua.edu.cn
jlxing@tsinghua.edu.cn, smxiang@nlpr.ia.ac.cn

## Abstract

Partial observability in environments poses significant challenges that impede the formation of effective policies in reinforcement learning. Prior research has shown that borrowing the complete state information can enhance sample efficiency. This strategy, however, frequently encounters unstable learning with high variance in practical applications due to the over-reliance on complete information. This paper introduces DCRL, a **D**ual **C**ritic **R**einforcement **L**earning framework designed to adaptively harness full-state information during training to reduce variance for optimized online performance. In particular, DCRL incorporates two distinct critics: an *oracle* critic with access to complete state information and a *standard* critic functioning within the partially observable context. It innovates a synergistic strategy to meld the strengths of the oracle critic for efficiency improvement and the standard critic for variance reduction, featuring a novel mechanism for seamless transition and weighting between them. We theoretically prove that DCRL mitigates the learning variance while maintaining unbiasedness. Extensive experimental analyses across the Box2D and Box3D environments have verified DCRL's superior performance.

## 1   Introduction

Real-world issues commonly involve a degree of partial observability, where agents lack access to the full system state and must base decisions on partial observations. These problems are typically defined as partially observable Markov decision processes (POMDPs) [1]. Despite the progress in reinforcement learning (RL) for addressing complex tasks [2–6], learning effective strategies in scenarios with partial observations remains challenging. Traditional RL methods in POMDP contexts often encounter impractical computations in processing the complete history, leading to suboptimal performance. Luckily, full states are accessible during training in many training scenarios. For example, in applications like autonomous driving[7] or robot control tasks [8], where computational and time limitations exist, only low-cost RGB images are available for deployment. At the same time, high-precision sensors can record more accurate and

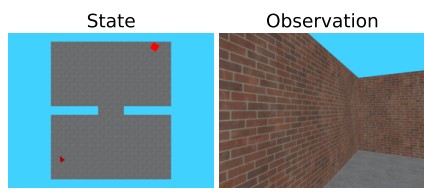

Figure 1: The WallGap environment, a procedurally generated navigation task in MiniWorld. The agent's goal (red triangle) is to reach the target (red box) within as few steps as possible under partial observability.

---

*Corresponding authors.

38th Conference on Neural Information Processing Systems (NeurIPS 2024).

detailed data during training. Similarly, in card games [9], the private hands of other players are typically hidden during deployment but revealed during training. This distinction of separating available information during training and deployment stages has driven the adoption of an offline learning and online execution paradigm in POMDP settings [10–13].

This study focuses on the asymmetric actor-critic framework [8, 14, 15], where the actor and critic rely on different inputs. The critic, serving as an auxiliary tool for training, can be transformed into an oracle critic to align with the offline learning and online execution paradigm. Specifically, the oracle critic is empowered to access the complete states while the actor remains constrained to partial observations. Using an oracle critic conditioned on the state and observation history promises to enhance performance and theoretically achieve unbiasedness [10]. However, the practical performance of such approaches raises doubts, particularly in domains characterized by high uncertainty and extensive state spaces. For instance, in the illustrated first-person 3D navigation task depicted in Figure 1, the agent is tasked with reaching a dynamic target that varies across episodes within as few steps as possible. This task involves a first-person view observation and an aerial view state. Introducing an oracle critic incorporating state information in this complex environment may increase variance, potentially undermining learning efficiency and overall performance.

To tackle the challenges mentioned above, we argue that the indiscriminate utilization of state information is inadvisable. We introduce a Dual Critic Reinforcement Learning framework (DCRL). DCRL consists of two critics: the *oracle* critic conditioned on the <history, state> tuple, and the *standard* critic conditioned solely on history. It introduces a synergistic strategy to meld the strengths of the oracle critic for efficiency improvement and the standard critic for variance reduction. In summary, the main contributions of this work are as follows:

- We propose a DCRL framework to address partially observable tasks. This framework features a novel mechanism for seamless transition and weighting between the dual critics.

- We theoretically prove that DCRL mitigates learning variance while maintaining unbiasedness.

- The DCRL framework is simple to implement. While our emphasis lies on the Advantage Actor-Critic (A2C) [16], DCRL can also be integrated into other actor-critic architectures, such as Proximal Policy Optimization (PPO) [17].

We extensively validate the DCRL framework based on A2C and PPO. Experimental results conducted on both Box2D and Box3D environments confirm its effectiveness. The source code is available in the supplementary material.

## 2 Related Work

Most methods can be adapted for partial observability by training memory-based policies to condition past observations. Recurrent Neural Networks (RNNs) [18] are commonly used for their simplicity and strong performance. Deep Recurrent Q-Networks (DRQN) [19] integrates Long Short-Term Memory (LSTM) [20] cells into the Deep Q-Network (DQN) [2] architecture, which demonstrated promising results. In addition to LSTM, other memory mechanisms such as Gated Recurrent Units (GRUs) [21, 22], Transformers [23, 24], and Language Models [25, 26] have been adopted in RL algorithms for memory learning. However, POMDPs are hard to solve when relying solely on memory-based policies because the size of the history grows linearly with the horizon length.

It is crucial that in many environments, complete states are observable during the training phase. Utilizing full observable states during training has emerged as a popular paradigm in RL. We discuss prior works on the utilization of full observability in RL algorithms. Additionally, we provide further discussion about training the world model, applications in multi-agent reinforcement learning, and other double structures in Appendix B.

**Using complete states to train the critic.** This topic is closely related to our paper and has been applied in many large-scale scenarios, such as DouDizhu [9] and Mahjong [27]. The critic in the actor-critic framework is a training tool not required during deployment, making its structure naturally suitable to exploit the state. This is achieved by critic asymmetry, where the actor and critic receive history and state, respectively. Asymmetric Actor-Critic [8] uses the critic conditioned on ground-truth states to address image-based robot control tasks. AACC [15] employs the critic conditioned on both states and observations to solve contextual Markov decision processes. Theoretically, using the

state as the critic input is biased, whereas using the <history, state> tuple is unbiased in expectation [10]. However, it introduces high variance in practice.

**Using complete states to train the policy.** These methods typically involve training a teacher policy using complete state information and then utilizing this policy to guide the learning of a student policy under partial observability. The forms of guidance vary, such as maintaining consistency of the Q-function [28], the encoder [11], action-distributions [29], and others. In addition to introducing extra loss terms to enforce consistency between the student policy and the teacher policy, the transition from the teacher policy to the student policy can also be facilitated through dropout of state information from the input [30–32]. Common issues include the imitation gap and suboptimal teacher, which can limit the potential capabilities of the student policy.

**Using complete states to learn the belief representation.** Belief state methods [1, 33] are classical in POMDPs. These methods often necessitate knowledge of the dynamics and reward functions, thus rendering them more applicable to scenarios with well-defined system dynamics. Consequently, they are typically constrained to simpler settings characterized by small state spaces, where the complexities of the environment are more tractable. Several recent studies [34–36] focus on learning belief representations by predicting future observations and rewards. VRM [34] utilizes a latent stochastic variable to model sequential observations. However, learning latent states that can capture long-term dependencies is challenging. Incorporating the complete states can help alleviate this issue. DouZero+ [37] introduces auxiliary tasks for predicting states while concurrently training policies conditioned on observations and the predicted states. Believer [13] decouples belief state modeling from policy optimization. They first capture state features through representation learning and then train the policy conditioned on observations and the predicted states.

## 3 Preliminaries

This section introduces the background topics required to understand our work: partially observable Markov decision processes, the recurrent Advantage Actor-Critic algorithm, and the Unbiased Asymmetric Actor-Critic algorithm.

### 3.1 Partially Observable Markov Decision Processes

A partially observable Markov decision process (POMDP) [1] is formally identified as a 7-tuple, namely $P = \langle \mathcal{S}, \mathcal{A}, T, \Omega, R, O, \gamma \rangle$, where $\mathcal{S}$, $\mathcal{A}$, and $\Omega$ represent the state space, action space, and observation space, respectively. The transition function $T : \mathcal{S} \times \mathcal{A} \to \Delta \mathcal{S}$ is denoted as $T(s, a, s') = p(s'|s, a)$, describing the probability of transitioning from state $s$ to state $s'$ after taking action $a$. The agent receives observations rather than states, and the observation function $O : \mathcal{S} \times \mathcal{A} \to \Delta \Omega$ is defined as $O(s, a, o)$ representing the probability of observing $o$ conditioned on state $s$ and action $a$. The reward function $R : \mathcal{S} \times \mathcal{A} \to \mathbb{R}$ is expressed as $R(s, a)$, representing the reward received by acting $a$ in state $s$, and $\gamma \in [0, 1]$ is the discount factor. In POMDPs, the optimal policy may depend on the entire history of observations and actions, denoted as $\mathcal{H} = \{h_t | h_t = (o_0, a_0, ...o_{t-1}, a_{t-1}, o_t)\}$. The belief-state $b : \mathcal{H} \to \Delta \mathcal{S}$ is the probability distribution over states $s$ given history $h$. The history reward function is defined as $R(h, a) = \mathbb{E}_{s|h}[R(s, a)]$. The agent aims to maximize the expected cumulative discounted rewards $\mathbb{E}[\sum_{t=0}^{\infty} \gamma^t R(s_t, a_t)]$ through interactions with the environment.

This paper assumes that both states $s_t$ and observations $o_t$ are accessible during training, while only the observations $o_t$ are accessible during deployment.

### 3.2 Recurrent Advantage Actor-Critic

Advantage Actor-Critic (A2C) [16] is primarily designed for fully observable control problems. It can be easily adapted to handle partially observable control problems by replacing all occurrences of the state $s$ with the respective history $h$. To clearly distinguish between different methods, we denote this method as **AC**. It trains an actor model $\pi : \mathcal{H} \to \Delta \mathcal{A}$ parameterized by $\theta$, and a critic model $V : \mathcal{H} \to \mathbb{R}$ parameterized by $\phi$. The policy loss is defined as:

$$\mathcal{L}_{\text{policy}}^{\text{AC}}(\theta) = -\mathbb{E}\Big[ \sum_{t=0}^{\infty} Q^\pi(h_t, a_t) \log \pi_\theta(a_t|h_t) \Big],  \tag{1}$$

where $Q^\pi(h_t, a_t)$ denotes the history-action value of the policy $\pi$. In practice, $Q^\pi(h_t, a_t)$ is replaced by the n-step bootstrapped advantage to reduce variance:

$$\delta_\phi(h_t, a_t) = \sum_{d=0}^{n-1} \gamma^d R(h_{t+d}, a_{t+d}) + \gamma^n V_\phi(h_{t+n}) - V_\phi(h_t). \tag{2}$$

Correspondingly, the policy loss is $\mathcal{L}_{\text{policy}}^{\text{AC}}(\theta) = -\mathbb{E}\big[\sum_{t=0}^{\infty} \delta_\phi(h_t, a_t) \log \pi_\theta(a_t|h_t)\big]$. In this work, the value loss is defined as $\mathcal{L}_{\text{value}}^{\text{AC}}(\theta) = \mathbb{E}\big[\sum_{t=0}^{\infty} \delta_\phi(h_t, a_t)^2\big]$. The entropy loss is $\mathcal{L}_{\text{entropy}}(\theta) = -\mathbb{E}\big[\sum_{t=0}^{\infty} \sum_{a\in\mathcal{A}} \pi_\theta(a_t|h_t) \log \pi_\theta(a_t|h_t)\big]$. The total loss is $\mathcal{L}_{\text{value}}^{\text{AC}}(\phi) + \mathcal{L}_{\text{policy}}^{\text{AC}}(\theta) + \eta \mathcal{L}_{\text{entropy}}(\theta)$, where $\eta$ is a weight for entropy regularization.

### 3.3 Unbiased Asymmetric Actor-Critic

Unbiased Asymmetric Actor-Critic [10] is a variant of recurrent A2C that can exploit state information during training. It considers an oracle critic model $V : \mathcal{H} \times \mathcal{S} \to \mathbb{R}$ parameterized by $\psi$ instead of the standard critic model. To clearly distinguish between different methods, we denote this method as **UAAC**. The n-step bootstrapped advantage for UAAC is defined as:

$$\delta_\psi(h_t, s_t, a_t) = \sum_{d=0}^{n-1} \gamma^d R(s_{t+d}, a_{t+d}) + \gamma^n V_\psi(h_{t+n}, s_{t+n}) - V_\psi(h_t, s_t). \tag{3}$$

Correspondingly, the policy model is trained to minimize the following objective function:

$$\mathcal{L}_{\text{policy}}^{\text{UAAC}}(\theta) = -\mathbb{E}\Big[\sum_{t=0}^{\infty} \delta_\psi(h_t, s_t, a_t) \log \pi_\theta(a_t|h_t)\Big]. \tag{4}$$

The value loss is defined as $\mathcal{L}_{\text{value}}^{\text{UAAC}}(\psi) = \mathbb{E}\big[\sum_{t=0}^{\infty} \delta_\psi(h_t, s_t, a_t)^2\big]$. Compared with the standard actor-critic method, the state $s$ provides additional context to determine the agent's true underlying situation, which may facilitate the learning of the oracle critic. Importantly, it has been proven that $V^\pi(h, s)$ provides Monte Carlo estimates of the respective $V^\pi(h)$ for arbitrary policies, which is theoretically unbiased, i.e.,

$$V^\pi(h) = \mathbb{E}_{s|h}[V^\pi(h, s)], \tag{5}$$
$$Q^\pi(h, a) = \mathbb{E}_{s|h}[Q^\pi(h, s, a)]. \tag{6}$$

Therefore, the equality $\nabla_\theta \mathcal{L}_{\text{policy}}^{\text{AC}}(\theta) = \nabla_\theta \mathcal{L}_{\text{policy}}^{\text{UAAC}}(\theta)$ holds, indicating that the policy gradient remains unchanged in expectation.

## 4 Dual Critic Reinforcement Learning

Critic models that learn accurate values slowly often become a bottleneck for policy performance in many scenarios. When state information is incorporated, it may be easier to learn $V^\pi(h, s)$ with a neural network compared with learning $V^\pi(h)$, leading to faster policy learning. However, $V^\pi(h, s)$ has nonzero variance. Taking into account the variance arising from $V^\pi(h, s)$: $\text{Var}_{s|h}[V^\pi(h, s)] = \mathbb{E}_{s|h}\big[\|V^\pi(h, s) - V^\pi(h)\|^2\big]$, with a practical batch size that is not too large, the variance tends to be high, especially in high-uncertainty belief states and large state spaces. In games like DouDizhu, when the initial information collection does not accurately infer the opponents' hands, training with $V^\pi(h, s)$ incorporating complete state information increases variance, thus limiting model generalization. Conversely, using $V^\pi(h)$, which relies solely on observations, enhances robustness in such cases. *This concept is akin to replacing the reward-to-go term with the Q-value to reduce variance in the vanilla policy gradient method, as discussed in [38, 39].*

### 4.1 Theories of the Dual Critic

**Definition 1.** Define the dual value function as $V_{\text{dual}}^\pi(h, s) = (1-\beta)V^\pi(h, s) + \beta V^\pi(h)$, where $\beta$ is a weight that balances the contributions of the $V^\pi(h, s)$ and $V^\pi(h)$, and $\beta \in [0, 1]$. Correspondingly, the dual Q function is defined as $Q_{\text{dual}}^\pi(h, s, a) = (1-\beta)Q^\pi(h, s, a) + \beta Q^\pi(h, a)$.

**Theorem 1.** *For arbitrary control problems and policies, $V_{\text{dual}}^{\pi}(h, s)$ is an unbiased estimate of $V^{\pi}(h)$ and $Q_{\text{dual}}^{\pi}(h, s, a)$ is an unbiased estimate of $Q^{\pi}(h, a)$.*

*Proof.* Following from Equation (5), we have:

$$\begin{aligned}
\mathbb{E}_{s|h}[V_{\text{dual}}^{\pi}(h, s)] &= \mathbb{E}_{s|h}\big[(1 - \beta)V^{\pi}(h, s) + \beta V^{\pi}(h)\big] \\
&= (1 - \beta)\mathbb{E}_{s|h}[V^{\pi}(h, s)] + \beta V^{\pi}(h) \\
&= V^{\pi}(h).
\end{aligned} \tag{7}$$

Similarly, as inferred from Equation (6), $\mathbb{E}_{s|h}[Q_{\text{dual}}^{\pi}(h, s, a)] = Q^{\pi}(h, a)$ by replacing the $V$ with $Q$ using the same approach outlined above. $\qquad\square$

Therefore, the variance of the dual value is defined as $\text{Var}_{s|h}[V_{\text{dual}}^{\pi}(h, s)] = \mathbb{E}_{s|h}\big[\|V_{\text{dual}}^{\pi}(h, s) - V^{\pi}(h)\|^2\big]$, and that of Unbiased Asymmetric Actor-Critic is defined as $\text{Var}_{s|h}[V^{\pi}(h, s)] = \mathbb{E}_{s|h}\big[\|V^{\pi}(h, s) - V^{\pi}(h)\|^2\big]$.

**Theorem 2.** *The variance of the dual value is smaller than that of the Unbiased Asymmetric Actor-Critic, i.e., $\text{Var}_{s|h}[V_{\text{dual}}^{\pi}(h, s)] \leq \text{Var}_{s|h}[V^{\pi}(h, s)]$.*

*Proof.* According to the definitions and $\beta \in [0, 1]$, it holds that

$$\begin{aligned}
\text{Var}_{s|h}[V_{\text{dual}}^{\pi}(h, s)] &= \text{Var}_{s|h}\big[(1 - \beta)V^{\pi}(h, s) + \beta V^{\pi}(h))\big] \\
&= (1 - \beta)^2 \text{Var}_{s|h}[V^{\pi}(h, s)] \\
&\leq \text{Var}_{s|h}[V^{\pi}(h, s)].
\end{aligned} \tag{8}$$

Similarly, $\text{Var}_{s|h}[Q_{\text{dual}}^{\pi}(h, s, a)] \leq \text{Var}_{s|h}[Q^{\pi}(h, s, a)]$ by replacing the $V$ with $Q$ using the same approach outlined above. $\qquad\square$

Considering the unbiasedness demonstrated above, we contemplate employing the dual Q function for policy updates. The policy gradient based on the dual Q function can be formulated as:

$$\nabla_\theta \mathcal{L}_{\text{policy}}^{\text{dual}}(\theta) = -\mathbb{E}\Big[\sum_t Q_{\text{dual}}^{\pi}(h_t, s_t, a_t)\nabla_\theta \log \pi_\theta(a_t|h_t)\Big]. \tag{9}$$

This formulation maintains the policy gradient unchanged in expectation.

**Theorem 3.** *The dual Q function maintains the policy gradient unchanged in expectation.*

*Proof.* Following Theorem 1, therefore,

$$\begin{aligned}
\nabla_\theta \mathcal{L}_{\text{policy}}^{AC}(\theta) &= -\mathbb{E}\Big[\sum_t Q^{\pi}(h_t, a_t)\nabla_\theta \log \pi_\theta(a_t|h_t)\Big] \\
&= -\sum_t E_{h_t, a_t}\Big[Q^{\pi}(h_t, a_t)\nabla_\theta \log \pi_\theta(a_t|h_t)\Big] \\
&= -\sum_t \mathbb{E}_{h_t, a_t}\Big[\mathbb{E}_{s_t|h_t}[Q_{\text{dual}}^{\pi}(h_t, s_t, a_t)]\nabla_\theta \log \pi_\theta(a_t|h_t)\Big] \\
&= -\sum_t \mathbb{E}_{h_t, s_t, a_t}\Big[Q_{\text{dual}}^{\pi}(h_t, s_t, a_t)\nabla_\theta \log \pi_\theta(a_t|h_t)\Big] \\
&= -\mathbb{E}\Big[\sum_t Q_{\text{dual}}^{\pi}(h_t, s_t, a_t)\nabla_\theta \log \pi_\theta(a_t|h_t)\Big] \\
&= \nabla_\theta \mathcal{L}_{\text{policy}}^{\text{dual}}(\theta).
\end{aligned} \tag{10}$$

$\square$

Similarly, the dual Q function decreases the variance of the policy gradient according to Theorem 2.

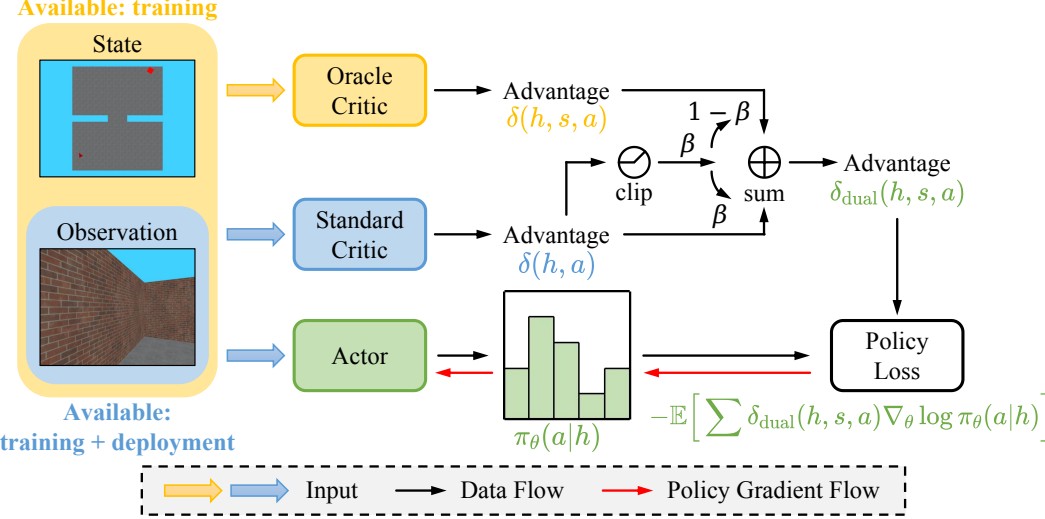

Figure 2: Implementation of DCRL framework. DCRL innovates a synergistic strategy to meld the strengths of the oracle critic for efficiency improvement and the standard critic for variance reduction.

## 4.2 Learning with Dual Critics

Building upon the abovementioned theory, we incorporate the dual Q function to balance variance and efficiency. This integration introduces the Dual Critic Reinforcement Learning framework (DCRL). As shown in Figure 2, the DCRL consists of three main components: 1) a *standard* critic network $V(h)$ parameterized by $\phi$, 2) an *oracle* critic network $V(h, s)$ parameterized by $\psi$, and 3) an actor model $\pi(a|h)$ parameterized by $\theta$. Inspired by the Equation (9) and Theorem 3, we propose the policy and value losses of DCRL as the following objective functions:

$$\mathcal{L}_{\text{policy}}^{\text{DCRL}}(\theta) = -\mathbb{E}\Big[\sum_{t=0}^{\infty}\big[(1-\beta)\delta_{\psi}(h_t, s_t, a_t)\text{log}\pi_{\theta}(a_t|h_t) + \beta\delta_{\phi}(h_t, a_t)\text{log}\pi_{\theta}(a_t|h_t)\big]\Big], \quad (11)$$

$$\mathcal{L}_{\text{value}}^{\text{DCRL}}(\phi, \psi) = \mathbb{E}\Big[\sum_{t=0}^{\infty}\delta_{\phi}(h_t, a_t)^2\Big] + \mathbb{E}\Big[\sum_{t=0}^{\infty}\delta_{\psi}(h_t, s_t, a_t)^2\Big], \quad (12)$$

where $\delta_{\phi}(h_t, a_t)$ and $\delta_{\psi}(h_t, s_t, a_t)$ follow the definitions from Equation (2) and Equation (3). The dual advantage is formulated as $\delta_{\text{dual}}(h_t, s_t, a_t) = (1-\beta)\delta_{\psi}(h_t, s_t, a_t) + \beta\delta_{\phi}(h_t, a_t)$.

We propose a novel weighting mechanism to synergistically meld the two critics and thoroughly exploit the complete state information. When the received return does not exceed the agent's value estimate, the advantage is non-positive ($\delta_{\phi}(h_t, a_t) \leq 0$), $\beta$ is clipped to 0. In cases with $\delta_{\phi}(h_t, a_t) > 0$, $\beta$ takes effect. $\beta$ is a parameter associated with $h$ but not with $s$. Hence, this mechanism maintains the theory mentioned above. This mechanism helps stabilize training and prevent large updates in cases where the advantage is non-positive. Similar to Self-Imitation Learning [40], this clipping mechanism can be viewed as a form of lower-bound-soft-Q-learning, where only samples with a positive advantage provide valuable insights into the optimal soft Q-value. From an empirical perspective, this approach embodies the notion that if there is still room for policy enhancement after being updated by the oracle critic $V(h, s)$, the standard critic $V(h)$ should be engaged. With full-state information from the oracle critic, DCRL avoids falling into a vicious cycle of poor policy and bad value estimates. The standard critic in DCRL can effectively mitigate variance, whereas recurrent Actor-Critic methods preclude such opportunities due to entanglement in a vicious cycle. These two critics establish a mutually supportive relationship, facilitating variance reduction while effectively using full-state information. The proof regarding lower-bound-soft-Q-learning is provided in Appendix C.

In addition to the mechanism mentioned above, there is another natural way to leverage the dual-critic framework by interchanging the roles of the oracle and standard critics. Specifically, we shift the weighting factor to the forefront of the $\delta_{\psi}(h_t, s_t, a_t)$ term which denote as $\beta'$, and the inside term

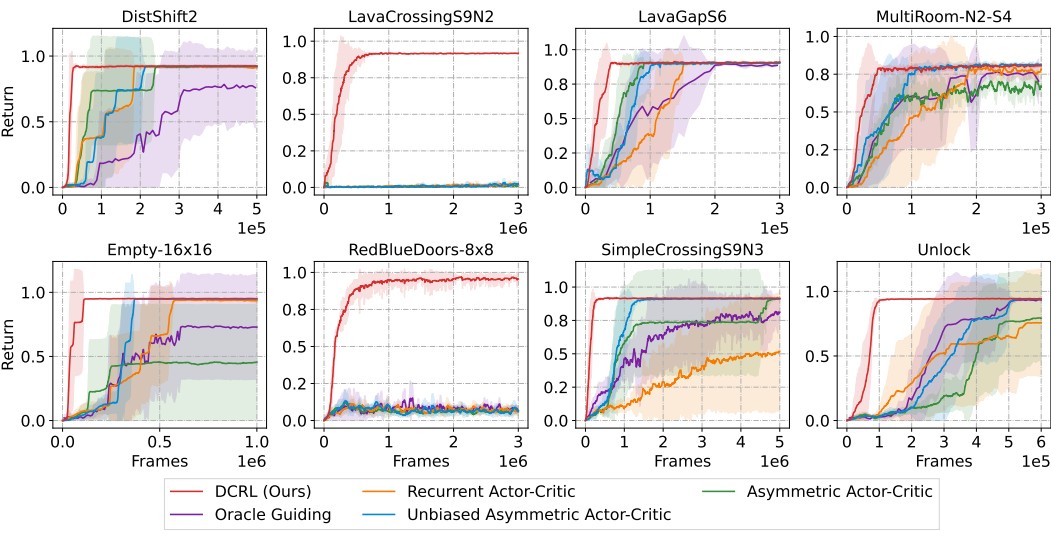

Figure 3: Learning curves on MiniGrid. The x-axis and y-axis represent the training frames and average returns, respectively. Shaded bars illustrate the standard error. All curves are trained based on A2C over 5 random seeds.

of the policy loss is defined as $(1 - \beta')\delta_\phi(h_t, a_t)\log\pi_\theta(a_t|h_t) + \beta'\delta_\psi(h_t, s_t, a_t)\log\pi_\theta(a_t|h_t)$. $\beta'$ is clipped to 0 when $\delta_\psi(h_t, s_t, a_t) \leq 0$ and otherwise takes a positive value. $\beta'$ is a parameter associated with both $h$ and $s$. Unlike DCRL, this alternative technique lacks the characteristic of unbiasedness and fails to thoroughly exploit state information while deviating from the lower-bound-soft-Q-learning framework. As a result, its performance exhibits instability compared with DCRL. The experimental section provides a detailed discussion of this technique's performance.

## 5 Experiments

We conduct a series of experiments to evaluate the effectiveness of DCRL in partially observable domains. Our research focuses on the model-free setting where complete states are available during training. We compare our DCRL approach with several baselines.

**Baselines.** Recurrent Actor-Critic is a recurrent version of the standard actor-critic algorithm, namely A2C [16] or PPO [17]. This baseline represents methods that do not utilize state information. Asymmetric Actor-Critic [8] is a representation method based on critic asymmetry. The critic uses the state as input, independent of the history. Oracle Guiding [30] is a representation method based on teacher policy. This approach gradually removes state information from the policy input, transitioning the teacher policy to a student policy under partial observability. The dropout probability linearly increases with training iterations until only observation remains. We adopt a 50% cutoff for all tasks. Unbiased Asymmetric Actor-Critic [10] is a state-of-the-art method for POMDPs. The critic uses the <history, state> tuple as input to exploit state information while introducing no bias into the training process, thus achieving superior performance. To ensure a fair comparison, we keep the main network architectures similar across all methods. We present the mean and standard deviation of results obtained from five different seeds for each environment. For a comprehensive overview of other implementation details and hyperparameters, please refer to Appendix E.

### 5.1 MiniGrid

To see how useful DCRL is across various POMDP tasks, we compare various methods in Box2D environments. MiniGrid [41] is a procedurally generated environment with goal-oriented tasks. As shown in Figure 4, the triangular agent with a discrete action space aims to reach the goal. These tasks involve interacting with different objects and solving diverse maze maps that vary in each episode. The observation comprises a first-person view representing a $7 \times 7 \times 3$ image. At the same time, the

state offers a fully observable perspective, with the image size varying according to the tasks. The image is compact and efficient encoding, rather than raw pixels.

All experiments are conducted based on A2C, and the results are plotted in Figure 3. Recurrent Actor-Critic struggles to learn a robust policy across most environments, primarily due to the agent's need for extensive exploration without state information. Despite leveraging states, Oracle Guiding and Asymmetric Actor-Critic performance exhibits instability, attributed to their heuristic nature and lack of theoretical guarantees. In contrast, the Unbiased Asymmetric Actor-Critic maintains stability and outperforms the mentioned methods. Nonetheless, it faces challenges in scenarios such as *LavaCrossingS9N2* and *RedBlueDoors-8x8*. Our DCRL framework surpasses all other methods in eight environments, particularly excelling in challenging tasks. Taking *LavaCrossingS9N2* and *Empty-16x16* as examples, the map size of the former is smaller, but the layout is more complex. In the case of

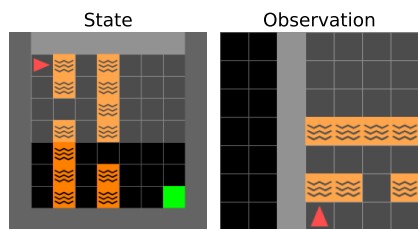

Figure 4: The LavaCrossingS9N2 environment in MiniGrid. The agent's goal (red triangle) is to reach the target (green box) within as few steps as possible under partial observability.

*Empty-16x16*, the states remain simple and relatively constant, making it easy to infer key information from history. Conversely, in *LavaCrossingS9N2*, objects change between episodes, leading to high uncertainty in the belief state. While leveraging states aids in precise value function learning, the variance poses challenges in policy optimization. DCRL effectively overcomes these limitations and emerges as a formidable competitor against other baseline methods.

Table 1: Performance of agents on MiniGrid after $1e5$ frames (27 games) and $1e6$ frames (13 games) of training. We report the mean and standard error of the performance.

| Methods | A2C | | PPO | |
|---|---|---|---|---|
| | $1e5$ frames | $1e6$ frames | $1e5$ frames | $1e6$ frames |
| Recurrent Actor-Critic | $0.32\pm0.13$ | $0.28\pm0.09$ | $0.26\pm0.07$ | $0.52\pm0.03$ |
| Asymmetric Actor-Critic | $0.26\pm0.10$ | $0.32\pm0.14$ | $0.27\pm0.08$ | $0.60\pm0.06$ |
| Oracle Guiding | $0.18\pm0.13$ | $0.27\pm0.06$ | $0.25\pm0.10$ | $0.48\pm0.05$ |
| Unbiased Asymmetric Actor-Critic | $0.31\pm0.08$ | $0.39\pm0.09$ | $0.28\pm0.10$ | $0.59\pm0.08$ |
| DCRL (Ours) | $\mathbf{0.63\pm0.07}$ | $\mathbf{0.89\pm0.01}$ | $\mathbf{0.47\pm0.08}$ | $\mathbf{0.84\pm0.03}$ |

To further validate the effectiveness of DCRL, we provide the visualization of different critic values in Appendix D.1. The visualization demonstrates that the dual value function $V_{\mathrm{dual}}(h, s)$ effectively reduces variance compared with the oracle critic. Besides, we evaluate our DCRL in 27 MiniGrid tasks based on A2C and PPO. The average score across all games is provided in Table 1 to demonstrate the overall performance for different methods. Our DCRL outperforms other baselines in 26 out of the 27 games. The detailed learning curves are available in Appendix D.2.

## 5.2 MiniWorld

This section investigates whether DCRL benefits more complex continuous state space tasks and whether it can be applied to other Actor-Critic algorithms, such as PPO. Unlike A2C, PPO lacks a strong theoretical connection to DCRL, but we claim they can still complement each other. To empirically verify this, we implement all methods based on PPO and evaluate them on the MiniWorld domain [41]. The observations and states are represented as $80 \times 60 \times 3$ images rendered as pixels. The results depicted in Figure 5 illustrate that DCRL enhances PPO performance across all four environments. In contrast to MiniGrid, the state format in this benchmark is not a concise encoding and tends to contain redundancy and noise. Although states assist the agent in effectively discerning information about the environment, the variance introduced by noise poses a challenge to the learning process. Furthermore, it is observed that the Unbiased Asymmetric Actor-Critic performs poorer than the Recurrent Actor-Critic across all tasks. Nevertheless, DCRL converges to superior policies by synergistically melding the two critics.

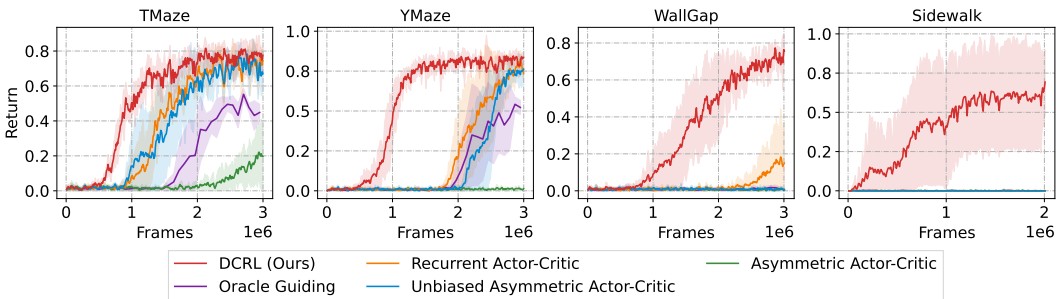

Figure 5: Learning curves on MiniWorld. The x-axis and y-axis represent the training frames and average returns, respectively. All curves are trained based on PPO over 5 random seeds.

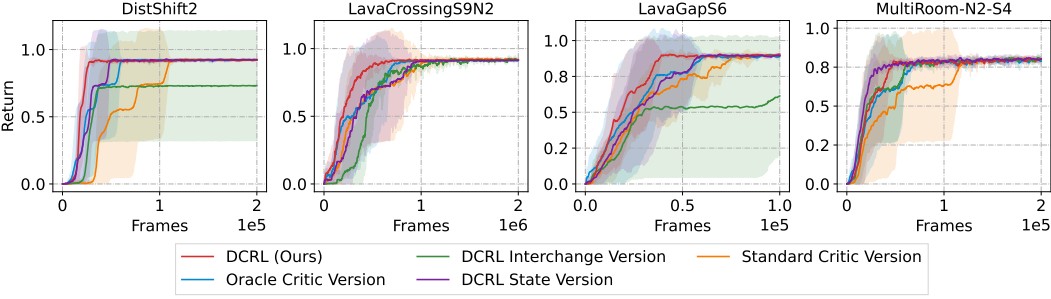

Figure 6: Ablation studies on MiniGrid to verify the two key factors of DCRL. All curves are trained based on A2C across 5 random seeds.

## 5.3 Ablation Studies

**Ablation analysis of components in DCRL.** We conduct detailed ablation studies to analyze how DCRL improves efficiency and enhances performance. DCRL contains two key factors: the dual-critic structure and the weighting mechanism. We replace the dual-critic structure with the single-critic structure to verify the structure's effectiveness. The Standard Critic Version and Oracle Critic Version are constructed with the following expressions for the inside terms of the policy loss: $\delta_\phi(h_t, a_t)\log\pi_\theta(a_t|h_t) + \beta\delta_\phi(h_t, a_t)\log\pi_\theta(a_t|h_t)$ and $\delta_\psi(h_t, s_t, a_t)\log\pi_\theta(a_t|h_t) + \beta'\delta_\psi(h_t, s_t, a_t)\log\pi_\theta(a_t|h_t)$. Next, we evaluate another weighting mechanism baseline (DCRL Interchange Version) discussed in Section 4.2, with the inside terms of the policy loss as $(1 - \beta')\delta_\phi(h_t, a_t)\log\pi_\theta(a_t|h_t) + \beta'\delta_\psi(h_t, s_t, a_t)\log\pi_\theta(a_t|h_t)$. The adjustment of $\beta$ and $\beta'$ follows the same way as discussed in Section 4.2. Additionally, we introduce an intuitive variant that modifies the *standard* critic's input from history to state, referred to as the DCRL State Version. This adjustment aligns with previous works that introduce biases; however, it has the potential to enhance efficiency gains. All methods utilize the same architectures and hyperparameters as DCRL.

We select four environments from the MiniGrid domain, where state information plays a crucial role. In these environments, methods that utilize states perform significantly better than those that do not. We compare our DCRL approach with four ablation baselines, and the results are presented in Figure 6. It can be observed that DCRL converges to optimal policies faster than other methods. Intuitively, incorporating the concept of positive advantage ($\delta_+$) can optimize the utilization of beneficial samples and expedite the training process. Consequently, single-critic structure methods integrating a $\delta_+$ term outperform those lacking it, as demonstrated by the superior performance of approaches like the Standard Critic Version compared with the Recurrent Actor-Critic. However, it is crucial that the two critics fulfill distinct and orthogonal roles within the dual-critic structure. DCRL incorporates an oracle critic that utilizes state information compared with the Standard Critic Version. The inclusion of state information provides the agents with additional context to accurately determine the environment's true state, facilitating learning an accurate value function and ultimately improving performance. In comparison to the Oracle Critic Version, DCRL reduces variance. These factors contribute to DCRL's efficiency. As for the DCRL Exchange Version, although it is also a dual-critic structure, its policy gradient is biased, and its weighting mechanism

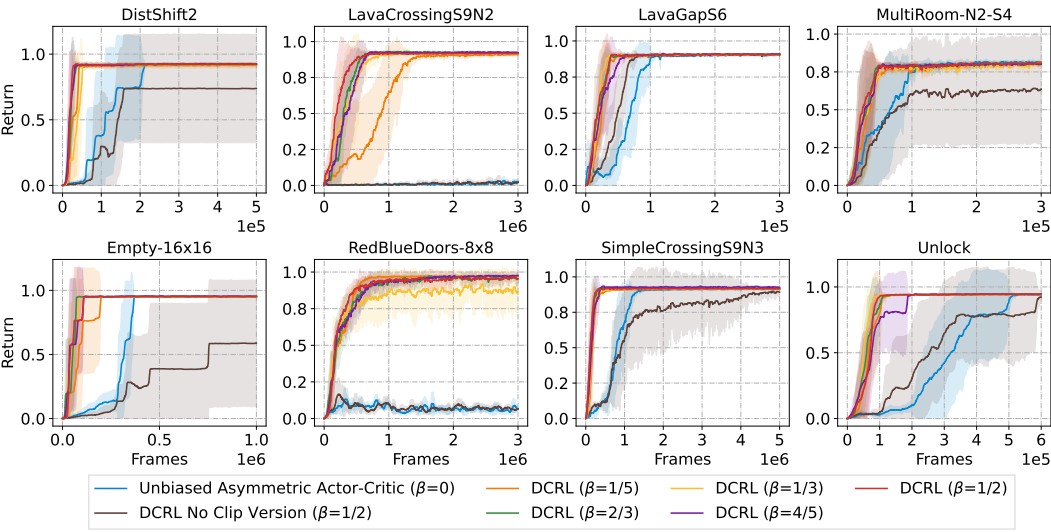

Figure 7: Ablation studies on MiniGrid to analyze the robustness of $\beta$ in DCRL. All curves are trained based on A2C across 5 random seeds.

does not utilize state information as effectively as DCRL. Its performance only shows advantages in limited environments. It converges to suboptimal policies in *DistShift2* and *LavaGapS6*. The DCRL State Version demonstrates commendable performance among the various DCRL variants. However, the biased characteristics of the DCRL State Version contribute to instability in its effectiveness. Notably, our DCRL achieves superior training performance in three out of four environments. These findings further highlight the significance of our DCRL framework.

**Ablation analysis of $\beta$ in DCRL.** Our DCRL incorporates a dynamic weighting mechanism, which can be interpreted as a form of lower-bound-soft-Q-learning, distinguishing it from conventional linear weighting methods. This mechanism substantially improves DCRL's performance while maintaining high stability concerning hyperparameters. To substantiate this claim, we implement a simplified weighting method with fixed ratios for weighting the two critics, referred to as the DCRL No Clip Version. Additionally, we conduct ablation experiments on the parameter $\beta \in \{1/5, 1/3, 1/2, 2/3, 4/5\}$. The results depicted in Figure 7 indicate that the DCRL No Clip version shows performance improvements in only some environments compared to the Unbiased Asymmetric Actor-Critic. In contrast, DCRL consistently enhances performance across all environments, underscoring the importance of the dynamic weighting mechanism. Furthermore, the stability of $\beta$ is notably evident. We choose $\beta = 1/2$ for reporting results in the paper, given its consistently superior performance.

# 6 Conclusion

This paper introduces a novel reinforcement learning framework designed for partially observable tasks. The motivation behind this work is that complete state information may be accessible during training in many scenarios. However, introducing a state could potentially lead to bias or variance. The proposed DCRL method leverages two asymmetric critics to use the state to expedite training and mitigate variance in practice. We underscore the benefits of training with two critics and provide theoretical evidence supporting the unbiasedness and variance reduction achieved through our approach. Empirical results obtained across multiple tasks serve to validate the efficacy of DCRL.

**Limitations.** Our framework operates under the assumption that state information is accessible during training. However, this assumption may not always hold, as only part of the state information might be available within a broader context. It is crucial to discuss whether this limitation will impact the performance of our framework, necessitating further investigation. Furthermore, our observations suggest that the availability of complete state information can sometimes hinder performance in certain environments. This observation underscores the need for a more nuanced approach to the design and utilization of state information. Addressing these issues could lead to enhancements in the performance and applicability of our framework.

## 7 Acknowledgement

This work was supported in part by the Natural Science Foundation of China under Grant No. 62222606 and 62076238.

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

## A Discussions

Incorporating state information can offer significant advantages for training, as it often encapsulates high-level features that improve the learning of $V(h, s)$ compared to $V(h)$ [10]. For example, in an extreme scenario where the state information $s$ represents the true value of $V(h)$, the agent can effortlessly infer this true value through $V(h, s)$. Specifically, in the context of the Empty Navigation task, when the state information indicates the agent's relative distance to the goal, the value function learning process is significantly streamlined. Besides, in most POMDPs, rewards are directly defined based on state information rather than on observations. Consequently, learning the value function $V(h, s)$ with the reward as the target is typically simpler than learning $V(h)$. This characteristic makes state information effective in most practical applications, a conclusion that numerous previous studies have empirically validated.

However, state information does not always enhance training. When the state information contains significant noise, introducing excessive noisy features to the input can hinder learning the value function. In unfamiliar environments, whether state information genuinely supports the learning process remains to be determined. The motivation behind this work is that state information may be accessible during training in many scenarios. However, introducing a state could potentially lead to bias or variance. In most cases, state information offers higher-level features that assist agents in decision-making. In these situations, DCRL enhances the process by further reducing variance. Conversely, when state information and partial observations correspond one-to-one, the oracle critic demonstrates no variance; DCRL may become ineffective under these circumstances.

## B Additional Related Work

**Using complete states to train the world model.** Traditional POMDP solutions rely on accurate environment models. Informed Dreamer [12] improves world modeling by incorporating state information. However, the dynamics models of POMDP environments are complex, posing challenges to modeling even with state information. Model inaccuracies will inevitably limit the performance. Our focus in this paper is on model-free methods.

**Using complete states in multi-agent reinforcement learning.** Centralized Training Decentralized Execution (CTDE) [42–44] is a prevalent paradigm for addressing challenges in multi-agent reinforcement learning (MARL). The optimization objective in MARL is to maximize the rewards under the global states, optimizing each agent accordingly. These methods mostly presuppose that the combination of optimized actions under partial observations leads to optimality under global states. Building on CTDE, MANSA[45] introduces a Global agent to decide when to activate global information usage. However, the policy that maximizes rewards under partial observations does not necessarily align with the policy that maximizes rewards under global states, i.e., $Q(h, a) \neq Q(s, a)$ [19]. Contrary to CTDE's assumptions, the learning objective of DCRL is to derive the optimal policy under partial observations.

**Double structure in the actor-critic framework.** In our work, we employ two asymmetric critics to address partially observable tasks. Previous studies have explored various double structures. In the context of double critics, Double DQN [46] utilizes a replicated Q-network as the target network to handle overestimation issues, a method later extended to subsequent works such as TD3 [47] and SAC [48]. Considering the opposite properties of single-critic and double-critic methods, TADD [49] incorporates a novel triplet critics mechanism to reduce the estimation bias. Additionally, ADC [50] takes advantage of environmental models by training a model-based critic, which is then combined with the model-free critic. This approach enables learning from the model-free path despite model noise, enhancing the robustness to environmental models. In the context of double actors, DARC [51] introduces a double actor structure to boost the agent's exploration ability by providing two paths for policy optimization, thus reducing the risk of the agent getting stuck locally. Moreover, DAC [52] utilizes a parallel actor-critic architecture to learn better options, indicating that the two critics can be effectively consolidated into one.

## C Additional Proofs

**Theorem 4.** *In the DCRL Interchange Version,* $\mathbb{E}_{s|h}\big[(1 - \beta')V^\pi(h) + \beta'V^\pi(h, s)\big] \neq V^\pi(h).$

*Proof.* Since $\beta'$ is a parameter associated with both $h$ and $s$, it cannot shift to the forefront of the expected symbol. Therefore, the dual value of the DCRL Interchange Version is biased. Also, the policy gradient in the DCRL Interchange Version is biased. $\square$

**Theorem 5.** *The learning objective* $\left(\delta_\phi(h,a)\right)_+ \log\pi(a|h)$ *in DCRL can be viewed as a form of lower-bound-soft-Q-learning.*

*Proof.* According to Self-Imitation Learning [40], the Lower bound soft Q-learning updates $Q(h,a)$ as follows:

$$L^{lb} = \mathbb{E}_{h,a,R\sim\mu}[\|\left(R - Q(h,a)\right)_+\|^2], \tag{13}$$

where $\mu$ is a behavior policy, $(h_t, a_t, R_t)$ is the trajectory triples from $\mu$, $R_t = r_t + \sum_{k=t+1}^{\infty} \gamma^{k-t}(r_k + \alpha\mathcal{H}_k^\mu)$ is the entropy-regularized return, $\alpha$ represents the weight of entropy bonus, and $\mathcal{H}_k^\mu = -\log\pi(a_k|h_k)$ is the entropy of the policy. According to the form of optimal soft value function and optimal policy in the entropy-regularized RL, it is natural to consider the following forms of a value function $V$ and a policy $\pi$:

$$V(h) = \alpha\log\sum_a \exp\left(Q(h,a)/\alpha\right), \tag{14}$$
$$\pi(a \mid h) = \exp\left((Q(h,a) - V(h))/\alpha\right), \tag{15}$$
$$Q(h,a) = V(h) + \alpha\log\pi(a \mid h). \tag{16}$$

For convenience, we can define the following:

$$\hat{R} = R - \alpha\log\pi(a \mid h), \tag{17}$$
$$\Delta = R - Q(h,a) = \hat{R} - V(h). \tag{18}$$

We can derive the gradient estimator of lower-bound-soft-Q-learning for the actor-critic architecture as follows:

$$
\begin{aligned}
&\nabla_\theta\mathbb{E}_{h,a,R\sim\mu}\left[\frac{1}{2}\left\|(R - Q(h,a))_+\right\|^2\right] \\
&= \mathbb{E}\left[-\nabla_\theta Q(h,a)\Delta_+\right] \\
&= \mathbb{E}\left[-\nabla_\theta\left(\alpha\log\pi(a \mid h) + V(h)\right)\Delta_+\right] \\
&= \mathbb{E}\left[-\alpha\nabla_\theta\log\pi(a \mid h)\Delta_+ - \nabla_\theta V(h)\Delta_+\right] \\
&= \mathbb{E}\left[\alpha\nabla_\theta\mathcal{L}_{\text{policy}}^{\text{lb}} - \nabla_\theta V(h)\Delta_+\right] \\
&= \mathbb{E}\left[\alpha\nabla_\theta\mathcal{L}_{\text{policy}}^{\text{lb}} - \nabla_\theta V(h)\left(R - Q(h,a)\right)_+\right] \\
&= \mathbb{E}\left[\alpha\nabla_\theta\mathcal{L}_{\text{policy}}^{\text{lb}} - \nabla_\theta V(h)\left(\hat{R} - V(h)\right)_+\right] \\
&= \mathbb{E}\left[\alpha\nabla_\theta\mathcal{L}_{\text{policy}}^{\text{lb}} + \nabla_\theta\frac{1}{2}\left\|\left(\hat{R} - V(h)\right)_+\right\|^2\right] \\
&= \mathbb{E}\left[\alpha\nabla_\theta\mathcal{L}_{\text{policy}}^{\text{lb}} + \nabla_\theta\mathcal{L}_{\text{value}}^{\text{lb}}\right],
\end{aligned}
\tag{19}
$$

where $\mathcal{L}_{\text{policy}}^{\text{lb}} = -\log\pi(a|h)\left(\hat{R} - V(h)\right)_+$ and $\mathcal{L}_{\text{value}}^{\text{lb}} = \frac{1}{2}\left\|(\hat{R} - V(h))_+\right\|^2$.

As $n\to\infty$, the learning objective in DCRL is $\left(\sum_{d=0}^{\infty}\gamma^d r_t - V(h)\right)_+ \log\pi(a|h)$. It can be viewed as a form of lower-bound-soft-Q-learning when $\alpha\to 0$. $\square$

In the DCRL Interchange Version, the equation $\mathbb{E}_{s|h}[\hat{R} - V(h,s)] = \hat{R} - V(h)$ holds according to Equation (5). However, the equation $\mathbb{E}_{s|h}[(\hat{R} - V(h,s))_+] = (\hat{R} - V(h))_+$ does not hold due to the presence of the $+$ operation. The inconsistency in the equation arises from the fact that the $+$ operation does not commute with the expectation operator, leading to a difference in the results. Consequently, the learning objective in the DCRL Interchange Version cannot be considered a form of lower-bound soft Q-learning.

# D Additional Experimental Results

## D.1 Visualization of Critic Values

To illustrate the effectiveness of DCRL in variance reduction, we visualize the values of different critics in the *Empty-Random-6x6* environment with a first-person view size of $3 \times 3$, as depicted in Figure 8. In this configuration, the same history is linked to different states. We collect 8 history samples, each associated with over 300 state samples, meaning that each $V^\pi(h)$ is connected to multiple $V^\pi(h, s)$ and multiple $V^\pi_{dual}(h, s)$. The converged critic models are used to predict $V^\pi(h)$, $V^\pi(h, s)$, and $V^\pi_{dual}(h, s)$, respectively. The 8 violin plots in Figure 9 depict the details of each history sample. These plots show density estimation of the values, with the center line indicating the mean value. It is evident that DCRL (shown in orange) demonstrates lower variance compared with the oracle critic (shown in blue).

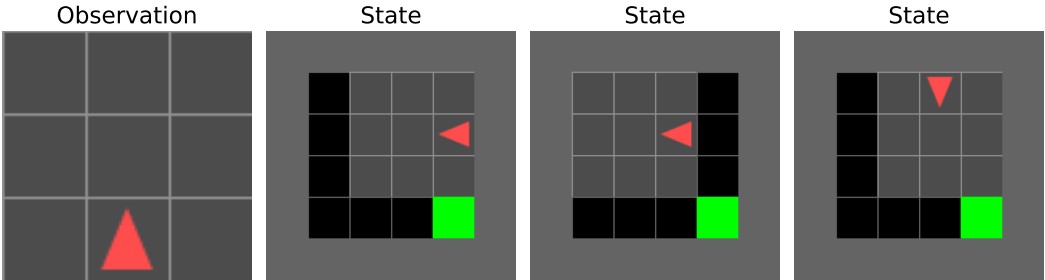

Figure 8: MiniGrid-Empty-Random-6x6. The dimensions of the first-person view are $3 \times 3$. The left figure illustrates the observation, while the right displays 3 different states that share the same observation.

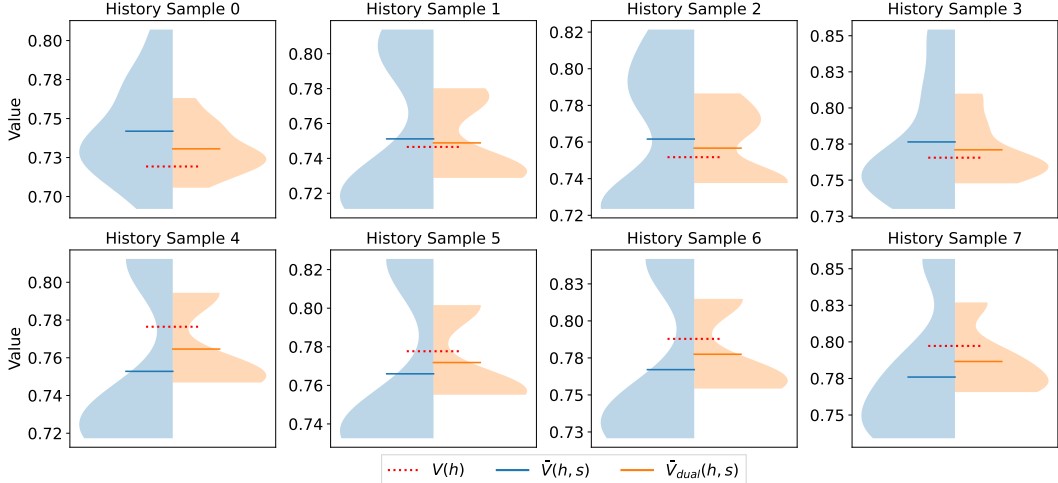

Figure 9: Visualization of 8 history samples. The violin plots show density estimation of the values, with the center line indicating the mean value.

## D.2 Learning Curves of All the Compared Methods

Figures 11 and 12 present the learning curves of all compared methods across 27 MiniGrid tasks, utilizing A2C and PPO, respectively.

## D.3 Learning Curves of Different $\beta$

Figure 13 presents the learning curves for different values of $\beta$ across 27 MiniGrid tasks.

## D.4 Comparison with DreamerV3

Although our research focuses on the model-free setting, we also compare DCRL with DreamerV3 [53], a widely used model-based method in POMDPs. DreamerV3 constructs a model of the environment and enhances its behavior through the imagination of future scenarios. Figure 10 presents the average scores over $1e^5$ steps across $8$ tasks in the MiniGrid environment. The results demonstrate that DCRL consistently outperforms DreamerV3 across all tasks. Notably, DCRL requires fewer computational resources and less training time than DreamerV3.

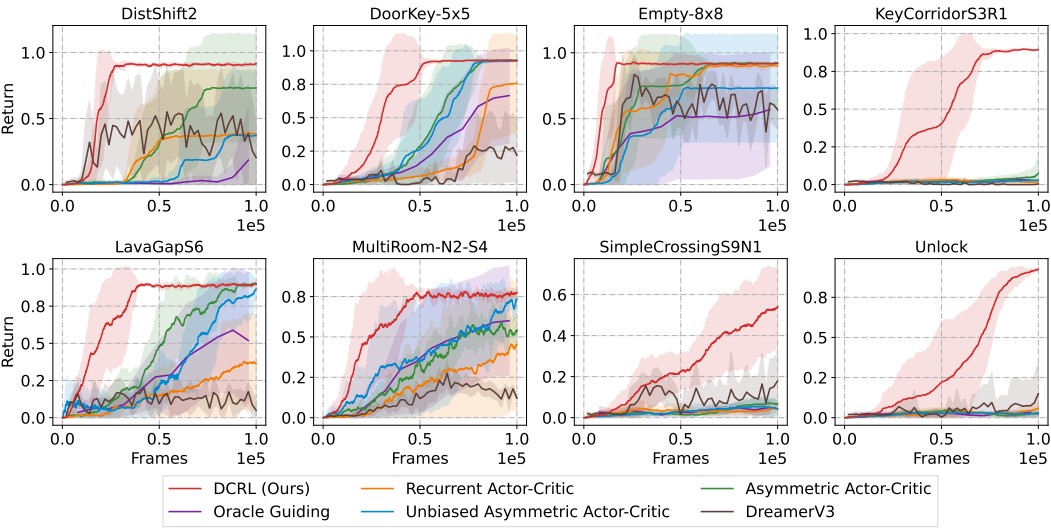

Figure 10: Learning curves on MiniGrid. All curves are trained based on A2C over $5$ random seeds.

# E Experiment Details

## E.1 Environment Descriptions

We describe each task environment used in our experiments. More detailed introductions can be found on the official project homepage of each benchmark. The source code is available in the supplementary.

***MiniGrid***
MiniGrid [41] comprises Box2D environments featuring various room layouts, interactive objects, and goals. A red triangular agent operates within a discrete action space, with actions such as "Move forward", "Turn left", "Turn right", "Pick up an object", "Toggle/activate an object", and others. Interactive objects include keys, doors, balls, and more. The agent must learn specific action sequences to achieve the final goal within its limited forward field of view, with rewards given only upon goal attainment. Each episode samples a different grid layout, and the agent has a first-person partially observable view of the environment.

We selected different experimental settings. For instance, as depicted in Figure 4, the agent in the *LavaCrossingS9N2* environment aims to reach the green goal square in the room's opposite corner while avoiding deadly lava rivers that terminate the episode in failure. Each lava stream runs either horizontally or vertically across the room, with a single safe crossing point. Fortunately, a path to the goal is guaranteed to exist.

The agent receives a $7 \times 7 \times 3$ dimensional image of the environment as observation. The three channels represent object ID, color ID, and state. The agent is positioned at the observation's bottom center, and the state is encoded similarly to the observation, consistently displaying all grids without adjusting to the agent's orientation. The final channel may include 1, 2, 3, or 4 values at the agent's current coordinates, indicating the agent's facing direction (east, south, west, and north).

*MiniWorld*

Similar to MiniGrid, MiniWorld [41] is a minimalistic Box3D interior environment simulator consisting of connected rooms with objects inside. In MiniWorld, the agent has a 3D partial view of the environment and must navigate to the red box. For simplicity, actions are discrete, including "Turn left", "Turn right", "Move forward", "Move back" and others. Turn and move actions rotate or move the agent by a small fixed interval, simulating the behavior of a differential drive robot. Rewards are based on the number of time steps required to successfully complete the task, with a small penalty incurred. If the task is not completed within the maximum steps, the current episode is terminated, and a reward of 0 is assigned.

We selected four experimental settings. For instance, as depicted in Figure 1, the *WallGap* environment features two rooms connected by a gap in a wall. The agent must navigate toward the red box, randomly placed in one of the rooms.

Observations consist of RGB images sized $80 \times 60 \times 3$ captured from the agents perspective, while states are RGB images of size $80 \times 60 \times 3$, representing an aerial view of the environment. In our experiments, RGB images are converted into grayscale images.

## E.2 Hyperparameters

We run all experiments on a single server with 64 Intel(R) Xeon(R) Gold 5218 CPU processors @ 2.30GHz and 1 Tesla V100 GPU. The hyperparameters for training each method are summarized in Table 2.

Table 2: Hyperparameters used for training each method.

| Hyperparameter | MiniGrid | MiniGrid | MiniWorld |
|---|---|---|---|
| Algorithm | A2C | PPO | PPO |
| Seeds in experiments | 5 | 5 | 5 |
| Discount factor $\gamma$ | 0.99 | 0.99 | 0.99 |
| $\lambda$ for GAE | 1 | 0.95 | 0.95 |
| Rollout steps | 5 | 512 | 512 |
| Number of workers | 16 | 16 | 16 |
| Entropy loss coef | 0.01 | 0.01 | 0.01 |
| Optimizer | RMSprop | Adam | Adam |
| learning rate | 1e-3 | 3e-4 | 3e-4 |
| max grad norm | 0.5 | 0.5 | 0.5 |
| PPO clip range | - | 0.2 | 0.2 |
| PPO training epochs | - | 4 | 4 |
| PPO mini-batch size | - | 512 | 512 |
| dual update per iteration | 16 | 4 | 4 |
| dual training epochs | 4 | 8 | 8 |
| dual batch size | 640 | 2048 | 2048 |
| $\beta$ | 0.5 | 0.5 | Best chosen from {0.1, 0.5} |

## E.3 Network Structures

In this section, we describe the structures of the policy network, the standard critic network, and the oracle critic network. The general architecture for all environments is the same, except for the representation of input images. We utilize the CNN model for representation, and the structures are contingent upon the dimensions $(n, m)$ of the input images. Detailed information for each environment is provided below.

**Representation Network**
*MiniGrid*
(1) $n \leq 3$ or $m \leq 3$
Conv2d(in=3, out=32, kernel=2, stride=1, pad=1),
Relu,
Conv2d(in=32, out=64, kernel=2, stride=1, pad=0),

Relu,
Conv2d(in=64, out=64, kernel=2, stride=1, pad=0),
Relu,
Flatten.
(2) $n \leq 15$ or $m \leq 15$
Conv2d(in=3, out=32, kernel=2, stride=1, pad=0),
Relu,
Conv2d(in=32, out=64, kernel=2, stride=1, pad=0),
Relu,
Conv2d(in=64, out=64, kernel=2, stride=1, pad=0),
Relu,
Flatten.
(3) $n > 15$ or $m > 15$
Conv2d(in=3, out=32, kernel=2, stride=1, pad=0),
Relu,
MaxPool2d(kernel=2),
Conv2d(in=32, out=64, kernel=2, stride=1, pad=0),
Relu,
Conv2d(in=64, out=64, kernel=2, stride=1, pad=0),
Relu,
Flatten.

*MiniWorld*
Conv2d(in=1, out=32, kernel=8, stride=4, pad=0),
Relu,
Conv2d(in=32, out=64, kernel=4, stride=2, pad=0),
Relu,
Conv2d(in=64, out=64, kernel=3, stride=1, pad=0),
Relu,
Flatten.

**Policy Network**
Representation Network,
FC(in=input_dim, out=64),
Relu,
LSTM(in=64, out=64).
FC(in=128, out=128),
Relu,
FC(in=128, out=number of actions).
**Standrad Critic Network**
The structure is the same as that of the policy network, except for the dimension of the final MLP output head, which is set to 1.
**Oracle Critic Network**
The structure is the same as that of the policy network, except for the dimension of the final MLP output head, which is set to 1.

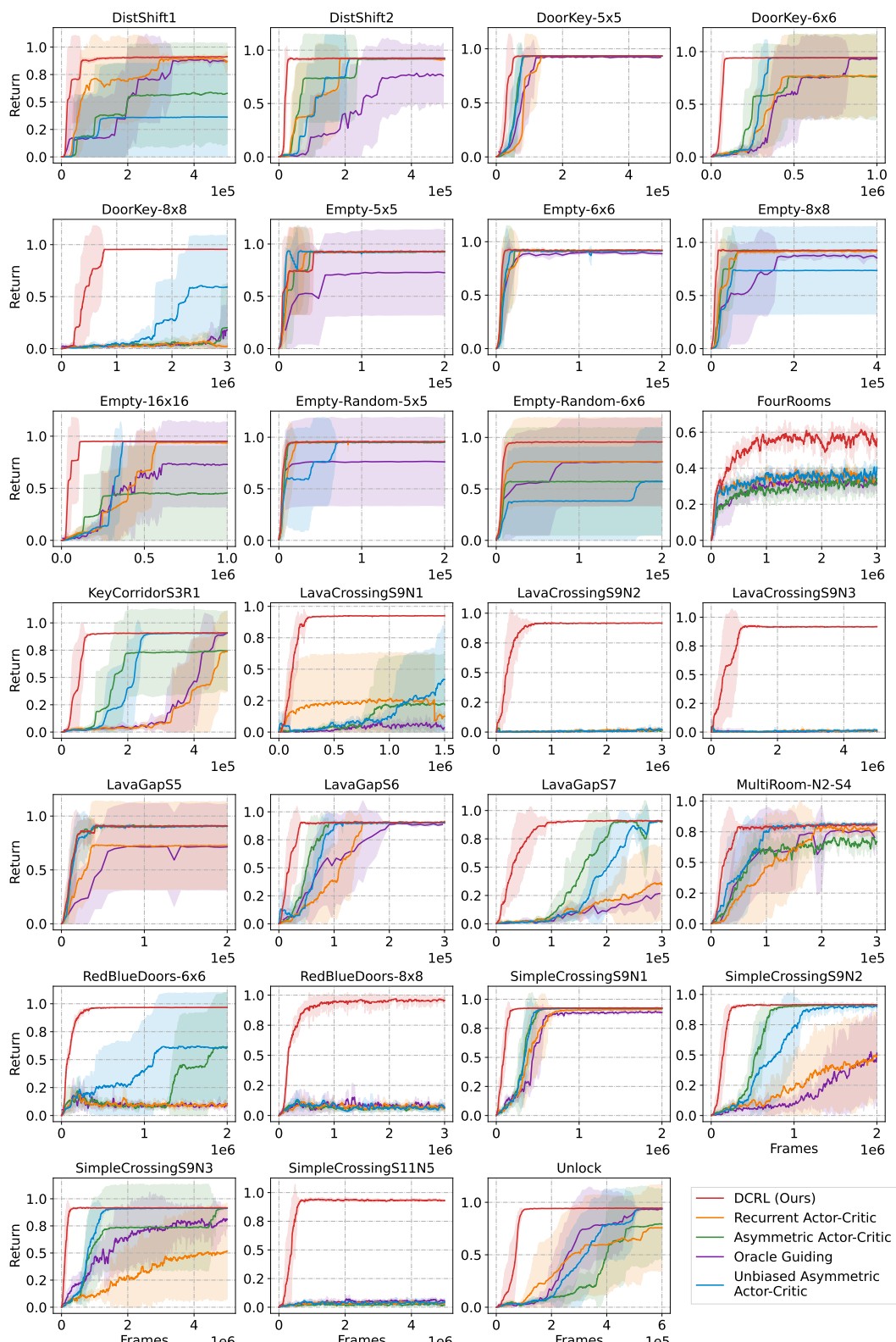

Figure 11: Learning curves on MiniGrid. All curves are trained based on A2C over 5 random seeds.

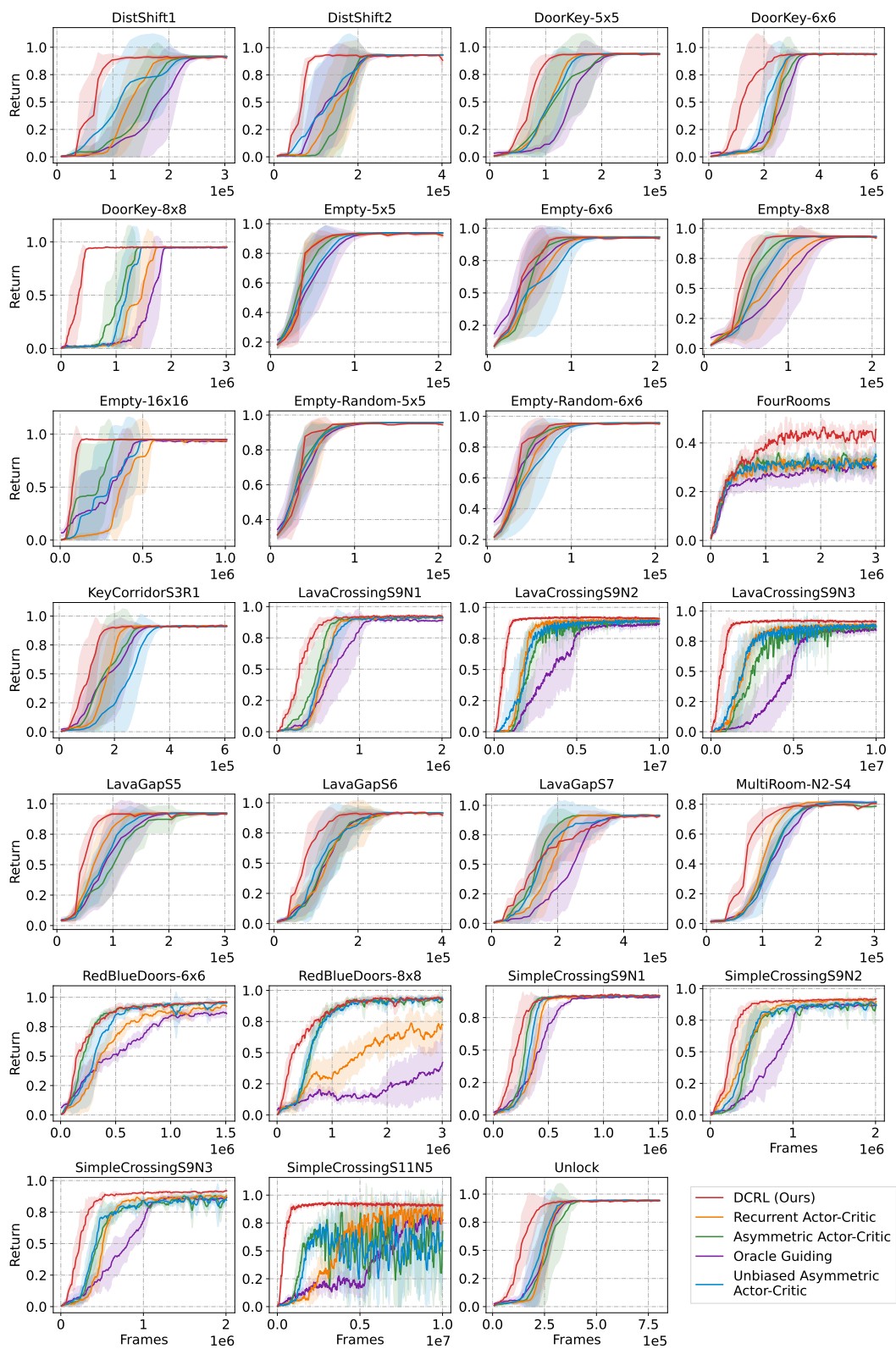

Figure 12: Learning curves on MiniGrid. All curves are trained based on PPO over 5 random seeds.

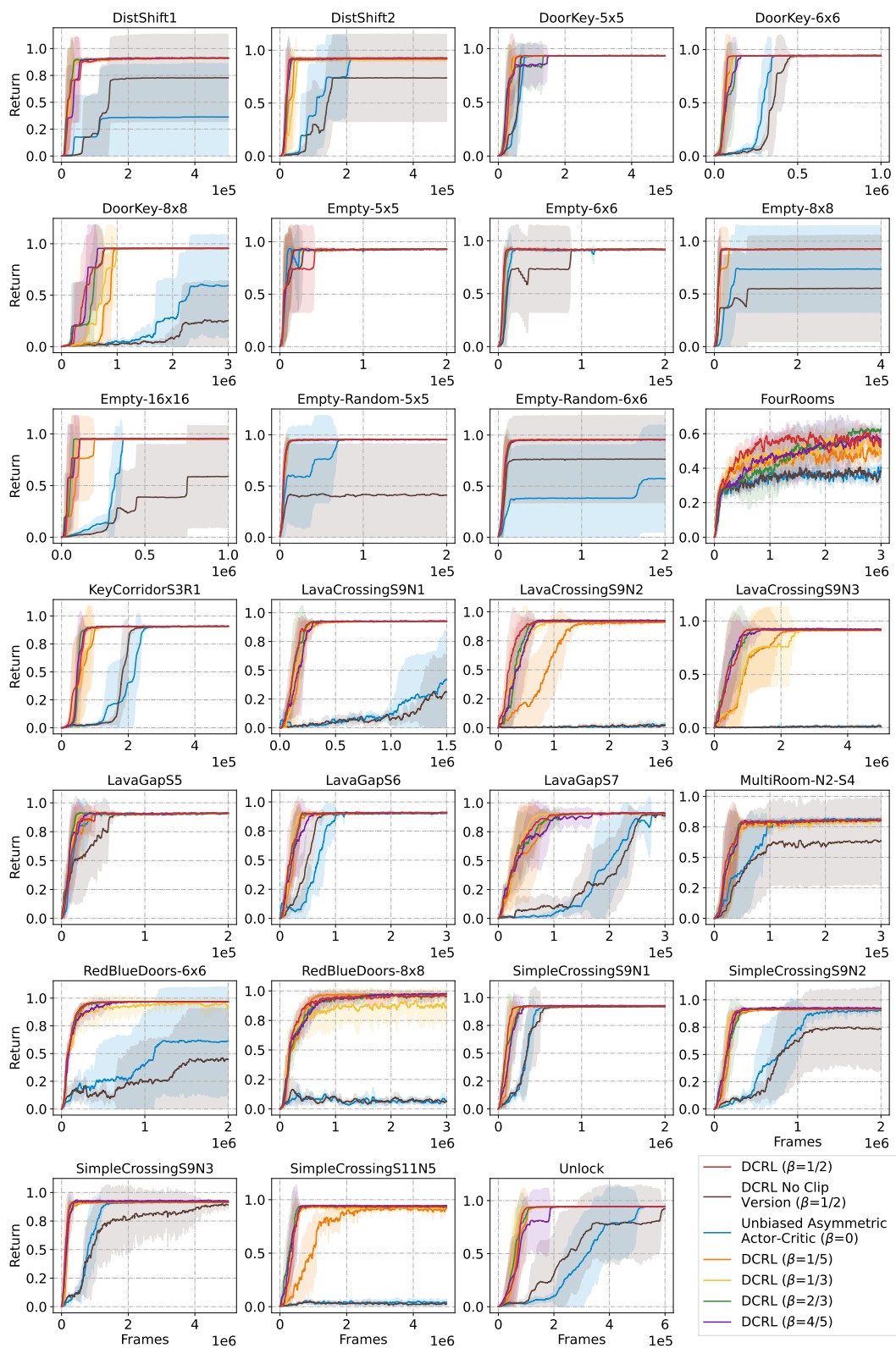

Figure 13: Ablation studies on different values of $\beta$ and DCRL No Clip Version. All curves are trained based on A2C over 5 random seeds.

