# OpenReview forum: "Dual Critic Reinforcement Learning under Partial Observability"
_NeurIPS.cc/2024/Conference — NeurIPS 2024 poster_

### Official Review · Reviewer_RNGm · 2024-06-28

**Soundness:** 2
**Presentation:** 2
**Contribution:** 2
**Rating:** 3
**Confidence:** 4

**Summary:**

This paper proposes a dual-critic architecture for the asymmetric RL regime where a policy deployable in partially observable settings is trained under full observability. The proposal is meant to be an improved version of Unbiased Asymmetric Actor Critic that, through its two critics, improves learning efficiency while reducing variance. This is demonstrated through theory and experiments in MiniGrid and MiniWorld.

**Strengths:**

**Originality**: The authors provide a new algorithm in the context of asymmetric RL. The contribution over related works is accurate and clearly stated.

**Quality**: The theory supporting the work appears sound. The experiments seem solid and are conducted across a number of partially observable domains, including both discrete gridworlds and continuous control environments. The results show that the method outperforms existing methods in terms of sample efficiency, sometimes by a substantial margin.

**Clarity**: The writing is mainly clear, particularly the motivation and the exposition of related works. A few technical details can be clarified however.

**Significance**: Deep RL for POMDP problems remains a significant challenge and new algorithms can have a substantial impact. Asymmetric RL has emerged as a promising paradigm for such problems and so I believe the work targets an important research topic.

**Weaknesses:**

**Originality**: Ultimately, the proposed method is somewhat derivative of two other well-known methods. The main contribution is the dual critic architecture, which is a linear combination of a typical recurrent critic $V(h)$ and the unbiased asymmetric critic $V(h,s)$. On its own, this would not be a major issue since the method is shown to perform well, but I believe the insights behind this approach are also discussed in the original Unbiased Asymmetric Actor Critic paper.

**Quality**: While the theorems appears correct, I wonder how relevant they are to the topic at hand. To my knowledge, what we *really* care about is the variance of the policy gradient Monte-Carlo estimator --- not the value estimator. The former is important since it plays a big role in the sample efficiency of policy learning. While the variance of the value estimator may indicate the amount of data required to train a reasonable value function, this is again only important insofar as reducing policy gradient variance through its role as a baseline.

There also needs to be a discussion on *why* we don't always want to reduce variance all the way to zero by setting $\beta = 1$ (which is equivalent to the simple recurrent critic). What value does increasing the variance (in order to incorporate state information) have? I believe this is not totally well understood in the literature either, but the topic should at least be brought up to clarify the point above. An investigation that can properly answer this question would be a strong contribution.

Regarding the implementation, it seems the choice of when to clip to $\beta = 0$ is very important but I don't understand why the particular heuristic in the paper was chosen. I'm not sure why the motivation of preventing large updates when the advantage is non-positive is relevant.

The results of the experiments show a solid advantage but it's not totally clear where the benefits are coming from (likely because I didn't understand the motivation behind the clipping of $\beta$). I think the experiments would also benefit from stronger baselines such as the Believer algorithm cited in the related work, and partially observable tasks that involve more than simple navigation.

**Clarity**: As previously mentioned, the reason why the state-based critic $V(h,s)$ is beneficial could be better explained, along with the heuristic for when to clip $\beta = 0$. Perhaps a toy example could be useful for this. I'm also unsure what the significance of the interchange method is.

**Significance**: I believe there isn't enough insight into the proposed approach to justify its use compared to Recurrent PPO or Unbiased Asymmetric Actor-Critic, which are somewhat better understood.

**Questions:**

1. Do you know how the dual critic architecture affects the variance of the policy gradient estimator?
2. How was the clipping method for $\beta$ chosen?

**Limitations:**

Yes.

---

> ### Author Rebuttal · Authors · 2024-08-05
>
> Thank you for your valuable feedback. We provide the following clarifications in response to your comments.
>
> > Weakness 1: About the insight.
> >
> > ... the proposed method is somewhat derivative of two other well-known methods. ... there isn't enough insight ...
>
> **In this study, we address the issue of high variance stemming from an over-reliance on state information. We demonstrate that state information is not always advantageous for training purposes.** In many environments, relying solely on partial observations can yield superior results compared to using state information.
>
> **Incorporating state information can offer significant advantages for training, as it often encapsulates high-level features that improve the learning of $V(h,s)$ compared to $V(h)$.** For example, in an extreme scenario where the state information $s$ represents the true value of $V(h)$, the agent can effortlessly infer this true value through $V(h,s)$. Specifically, in the Empty Navigation task, when state information indicates the agent's relative distance to the goal, the value function learning process is significantly streamlined. In addition, the reward function is defined based on the state rather than the observation. Consequently, learning for the standard critic necessitates implicit consideration of the distribution $b(s|h)$. In contrast, **the oracle critic bypasses this requirement by directly learning the reward**, as the distribution is explicitly provided through the sampling operation during the policy gradient calculation.
>
> **However, state information is not always advantageous for training.** When the state information contains significant noise, **adding excessive noisy features to the input can hinder learning the value function.** In unfamiliar environments, whether state information genuinely supports the learning process remains to be determined.
>
> To address this issue, we introduce a DCRL framework that leverages the strengths of two critics. **We provide theoretical evidence indicating that DCRL mitigates the learning variance while maintaining unbiasedness. Moreover, our approach incorporates a dynamic weighting mechanism, which can be interpreted as a form of lower-bound-soft-Q-learning, distinguishing it from conventional linear weighting methods.** Compared to Recurrent AC and UAAC methods, our proposed DCRL demonstrates substantial improvements.
>
> > Weakness 2 and Question 1: About the variance reduction in policy gradient.
> >
> > ... the variance of the policy gradient Monte-Carlo estimator ... how the dual critic architecture affects the variance of the policy gradient estimator?
>
> We fully understand your concerns regarding the variance of the policy gradient Monte Carlo estimator. Traditional Actor-Critic methods employ the reward-to-go term to compute the policy gradient. Nonetheless, **the variance of the reward-to-go tends to be high**, which has led researchers to substitute this term with the Q-value to mitigate variance in the standard policy gradient method, i.e., $Q(h_t, a_t) \nabla_{\theta}\log\pi_{\theta}(a_{t}|h_{t})$. Similarly, in our study, we focus on the policy $\pi(a|h)$, which is linked to **a unique $Q(h, a)$ but can also correspond to multiple $Q(h, s, a)$ values** due to the non-one-to-one relationship between state and history. **Consequently, the variance of the policy gradient derived from $Q(h, a)$ is lower than that obtained from $Q(h, s, a)$.** As stated in Theorem 1 of our paper, $Q_{\mathrm{dual}}(h, s, a)$ in DCRL achieves a reduction in this variance compared to $Q(h, s, a)$ by integrating the standard critic, thereby decreasing the variance of the policy gradient.
>
> > Weakness 3 and Question 2: About the $\beta$.
> >
> > ... reduce variance all the way to zero by setting $\beta=1$ ... when to clip $\beta=0$ ... the motivation behind the clipping of $\beta$ ... How was the clipping method for $\beta$ chosen?
>
> In response to weakness 1, we elaborated on the advantages of employing the oracle critic $V(h, s)$, which incorporates state information for policy training.
>
> The motivation behind the clipping mechanism is to leverage the standard critic when opportunities for policy enhancement remain following updates from the oracle critic. **This approach can be interpreted as a form of lower-bound soft Q-learning, wherein only samples exhibiting a positive advantage contribute meaningful insights into the optimal soft Q-value.** To validate this concept, we implemented a simplified weighting method that employs fixed ratios to weight the two critics (referred to as DCRL No Clip Version). As illustrated in Figure 1 of the rebuttal PDF, this method shows performance improvements in only certain environments, confirming that the dynamic weighting mechanism of DCRL is crucial.
>
> To evaluate the impact of the parameter $\beta$, we conducted ablation experiments testing various values of $\beta \in \\{1/5, 1/3, 1/2, 2/3, 4/5\\}$. **The choice of $\beta$ demonstrates notable stability.** **This stability is primarily attributable to the characteristics inherent in lower-bound soft Q-learning.**
>
> > Weakness 4:
> >
> > ... the experiments would also benefit from stronger baselines...
>
> Thank you for this valuable suggestion. We have followed your advice and considered the commonly used POMDP baseline DreamerV3.  We compared the performance of DCRL, DreamerV3, and additional baselines across eight tasks in the MiniGrid environment. Figure 3 of the rebuttal PDF illustrates the average scores from five training seeds over $1e5$ steps for each method. **The results indicate that DCRL consistently outperforms DreamerV3 across all eight tasks**. Importantly, DCRL requires fewer computational resources and less training time than DreamerV3.
>
> Thank you for your insightful and thoughtful reviews, which have significantly improved the quality of our paper. If you find that these concerns have been resolved, we would appreciate it if you would reconsider your ratings of our paper.

---

> ### Author Response · Authors · 2024-08-12
> **Have we addressed your concerns?**
>
> Thanks again for your time and effort in reviewing our paper! As the discussion period is coming to a close, we would like to know if we have resolved your concerns expressed in the original reviews. We remain open to further feedback and are committed to implementing additional improvements if necessary. If you find that these concerns have been resolved, we would appreciate it if you would consider reflecting this in your rating of our paper!

---

> > ### Comment · Reviewer_RNGm · 2024-08-13
> >
> > Thank you to the authors for clarifying my questions.
> >
> > I appreciate your clarification on the connection between the policy gradient variance and the value function variance. The paper should better distinguish these two concepts -- "variance" on its own is typically associated with the policy gradient estimator and it's important that these concepts not be conflated. Your explanation of how they are mutually related helps allay my concerns.
> >
> > I also appreciate the insight behind why we would sometimes opt for the higher variance $V(h, s)$ critic. **However, I'm not sure I'm convinced by the explanation of the dynamic weighting mechanism.** As I understand it, there are some situations where we want the $V(h, s)$ critic to have higher weight and others when we want $V(h)$ to have higher weight -- presumably $V(h)$ ought to be weighted higher near the end of training when we've converged on a good $V(h)$ critic and $V(h, s)$ unnecessarily adds (irreducible) sampling variance. The argument that the DCRL critic has lower variance and is unbiased compared to $V(h)$ is misleading since that would suggest we always want to choose $\beta = 1$. In reality, the question of *when* each critic is more effective is the key question here.
> >
> > That leads me to my next point. The authors implicitly claim that the dynamic weighting is an effective way to decide when to apply the DCRL critic vs the $V(h)$ critic. To me, the explanation in the paper (and in the rebuttal) is a bit brief and lacking -- it still doesn't quite answer why this works so well. Why is the sign of the advantage $\delta_\phi(h_t, a_t)$ a good determinant of how we should weight the two critics?
> >
> > Lastly I appreciate the inclusion of Dreamer-v3 as a baseline that does not utilize state information. I still think including additional baselines that do leverage state information would be an improvement.

---

> > > ### Author Response · Authors · 2024-08-13
> > >
> > > Thank you very much for your response. We are happy to hear that we have addressed some of your concerns. We will incorporate your suggestions regarding the concepts of variance and provide a more precise description in the revised manuscript. For your remaining questions, we provide point-to-point responses as follows:
> > >
> > > > Q1: About the dynamic weighting mechanism.
> > >
> > > We apologize for the misunderstandings that have arisen. To clarify, our claim is that DCRL reduces variance compared to UAAC (which relies solely on $V(h,s)$) and is unbiased with respect to $V(h)$ in expectation, leading to an unbiased policy gradient. **However, this does not imply that we always want to choose $\beta=1$.**  When $\beta=1$, state information can not be utilized. In fact, $\beta=1$ is not used in either the derivations or experiments of DCRL. As shown in Figure 1 of the rebuttal PDF, we tested values of $\beta$ in $\\{1/5, 1/3, 1/2, 2/3, 4/5\\}$ and selected $\beta=1/2$ for reporting our results in the paper.
> > >
> > > We agree that the key question is when each critic is more effective. Our motivation is to enhance the balance between the two critics. DCRL achieves this by dynamically adjusting the weight rather than always setting $\beta$ to $1$ or $0$. Specifically, the adjustment mechanism is based on the advantage $\delta(h, a)$. This approach aims to reduce variance while preserving the acceleration provided by state information. From the perspective of lower-bound-soft-Q-learning, **the policy gradient induced by $V(h)$ supports the lower bound of the optimal Q-values only when the advantage is positive.** This allows the policy to update towards higher returns without altering the update direction from $V(h, s)$. Conversely, when the advantage is non-positive, $V(h)$ does not provide useful information about the optimal Q-values and may interfere with updates from $V(h, s)$, thus diminishing the benefit of state information. Therefore, $V(h)$ is excluded from training in DCRL under these conditions. As shown in Figure 1 of the rebuttal PDF, the results of the DCRL No Clip Version experiment support this conclusion.
> > >
> > > We also agree that $V(h)$ ought to be weighted higher near the end of training.  However, this presents a challenge: accurately identifying the later stages of training in an unfamiliar environment can be difficult. Mismanagement of this weighting may lead to suboptimal solutions. Therefore, we adopt a more intuitive and straightforward approach: **applying $V(h)$ whenever it reduces variance without disrupting updates of state information.** The idea that $V(h)$ ought to be weighted higher near the end of training is the direction of our future efforts.
> > >
> > > > Q2: About additional baselines that leverage state information.
> > >
> > > Thank you for your valuable feedback. Our paper has included three relevant latest baselines that utilize state information: UAAC, AAC, and Oracle Guiding. Following your insightful suggestions, we are conducting experiments about Believer. As Believer focuses on representation models instead of learning frameworks like DCRL, it involves collecting random samples to pre-train the representation model for $5000$ epochs, which considerably extends training time. We will consider incorporating additional baselines in the revised version of our manuscript as per your recommendation.
> > >
> > > We sincerely appreciate the time and effort you have invested to the discussions! Please feel free to let us know if you have any further concerns.

---

### Official Review · Reviewer_VBT3 · 2024-07-08

**Soundness:** 3
**Presentation:** 4
**Contribution:** 2
**Rating:** 5
**Confidence:** 4

**Summary:**

The authors propose a methood that uses a weighted dual critic structure for tackling POMDPs. The dual critic structure consists of a critic that receives global state information while the other critic receives only the partially observations of the state. The authors provides some simple yet concrete analytical results that show the method induces unbiased critic estimates and can reduce variance compared to that of the Unbiased Asymmetric Actor-Critic.

**Strengths:**

The paper is well written and easy to follow. The motivation for their approach is well laid out and although the focus of the paper is to provide a new methodology which is supported by experiments, the paper includes some lite-touch analytics that support the overall ideas well.

The idea is quite simple and intutive, the presentation is nevertheless very clean with the authors having laid out both the rationale and benefits for the method in a very clear way.

**Weaknesses:**

1. While I understand the reason for doing so, such a strategy introduces a harsh jump-discontinuity in the dual advantage and critic - this may lead to numerical instabilities. The paper would benefit from a discussion on this issue.

2. The method also seems related to centralised training and decentralised execution in multi-agent reinforcement learning. There, the critic is trained with global information while the actor is trained using only local inputs. It would be a nice connection to make if the authors could include some discussion on degenerate case of ct-de with N=1.

3. Notwithstanding the discussions on why the method performs well in the chosen environments (lines 297-314) I would have liked to have seen some discussion and analysis of when one could expect this method to be most useful. Indeed, it's conceivable that including more information may slow training even if one can expect improved asymptotic performance.  There may also be situations where we do not have access to the full global state even during training.  I think the paper would benefit from such discussions.

**Questions:**

Q1. The authors state that $V^\pi(h)$ has nonzero variance. Using the definition of the dual value function, setting $\beta\equiv 1$ implies that $V_{\rm dual}^\pi(h,s)=V^\pi(h)$. However, setting $\beta=1$ in equation 8 implies that $Var_{s|h}[V_{\rm dual}^\pi(h,s)]=0$. Can the authors explain this.

Q2. How does the method perform over a range of values of $\beta$?

Q3. What environments should we expect the method to do well in (and not so well)?

**Limitations:**

L1.The experiment section does not contain an ablation on the value of $\beta$. Without this it is difficult to understand the size of the range of values of $\beta$ for which the method performs well.


L2. The experiment environments are quite simple - it would be useful to see how the method performs on complex benchmark environments.

---

> ### Author Rebuttal · Authors · 2024-08-05
>
> Thank you for your constructive comments and suggestions. Below, we provide detailed responses to each point.
>
> > Weakness 1:
> >
> > ... harsh jump-discontinuity ... numerical instabilities.
>
> We fully understand your concerns. **The numerical instability arises primarily from the variance of $V(h, s)$ due to the expectation of $V(h, s)$ is equal to $V(h)$.** Figure 8 in the appendix illustrates that our DCRL demonstrates a more stable value distribution and experiences fewer numerical instabilities than UAAC, attributable to its reduced variance.
>
> > Weakness 2:
> >
> > ... degenerate case of ct-de with N=1.
>
> We appreciate the valuable insights you have provided. The degenerate case of CTDE when $N=1$ corresponds to a standard POMDP. In this context, the well-known multi-agent actor-critic algorithm COMA, utilizes both state information ($s$) and history ($h$) as inputs for the critic during centralized training. Conversely, during execution, the actor relies solely on history as its input. **This situation can be interpreted as a specific instance of the UAAC framework.** We have incorporated relevant discussions on this topic in the revised manuscript.
>
> > Weakness 3.1 and Question 3:
> >
> > ... when one could expect this method to be most useful.
> >
> > What environments should we expect the method to do well in (and not so well)?
>
> **Incorporating state information can offer significant advantages for training, as it often encapsulates high-level features that improve the learning of $V(h,s)$ compared to $V(h)$.** For example, in an extreme scenario where the state information $s$ represents the true value of $V(h)$, the agent can effortlessly infer this true value through $V(h,s)$. Specifically, in the context of the Empty Navigation task, when the state information indicates the agent's relative distance to the goal, the value function learning process is significantly streamlined. Furthermore, the reward function is defined based on the state rather than the observation. Consequently, the standard critic's learning necessitates implicit consideration of the distribution $b(s|h)$. In contrast, **the oracle critic bypasses this requirement by directly learning the reward**, as the distribution is explicitly provided through the sampling operation during the policy gradient calculation.
>
> **However, state information does not always enhance training.** When the state information contains significant noise, **introducing excessive noisy features to the input can hinder learning the value function**. In unfamiliar environments, whether state information genuinely supports the learning process remains to be determined.
>
> The motivation behind this work is that state information may be accessible during training in many scenarios. However, introducing a state could potentially lead to bias or variance. **In most cases, state information offers higher-level features that assist agents in decision-making. In these situations, DCRL enhances the process by further reducing variance. Conversely, when state information and partial observations correspond one-to-one, the oracle critic demonstrates no variance; DCRL may become ineffective under these circumstances.**
>
> > Weakness 3.2
> >
> > ... more information may slow training ... do not have access to the full global state ...
>
> The scope of our study encompasses situations where state information is accessible during training. This assumption is prevalent in previous works and reflects real-life common scenarios.
>
> We provide the runtime in seconds for each method below regarding the potential issue of increased training duration due to additional information. DCRL requires more time to execute compared to other baselines, primarily because it trains an additional critic network. Nevertheless, **DCRL demonstrates significantly higher sample efficiency and performs better with fewer environment steps, thus compensating for the slower wall time.**
>
> Table 1: Comparison of DCRL and other baselines in MiniGrid. The first number represents the mean ± std of performance, while the second number indicates the runtime in seconds.
>
> |                                  |   A2C ($1e5$ frames)   |   A2C ($1e6$ frames)    |   PPO ($1e5$ frames)   |   PPO ($1e6$ frames)    |
> | :------------------------------: | :--------------------: | :---------------------: | :--------------------: | :---------------------: |
> |      Recurrent Actor-Critic      |   0.32±0.13, 108.41s   |   0.28±0.09, 1026.31s   |   0.26±0.07, 111.90s   |   0.52±0.03, 1130.45s   |
> |     Asymmetric Actor-Critic      |   0.26±0.10, 106.49s   |   0.32±0.14, 1000.69s   |   0.27±0.08, 110.21s   |   0.60±0.06, 1107.77s   |
> | Unbiased Asymmetric Actor-Critic |   0.31±0.08, 108.89s   |   0.39±0.09, 1066.49s   |   0.28±0.10, 114.35s   |   0.59±0.08, 1160.74s   |
> |         **DCRL (Ours)**          | **0.63±0.07, 130.12s** | **0.89±0.01, 1224.29s** | **0.47±0.08, 132.70s** | **0.84±0.03, 1331.69s** |
>
> > Question 1:
> >
> > ... $V^{\pi}(h)$ has nonzero variance.
>
> We apologize for the misunderstanding caused by a typo in our paper. In line 155, we clarify that $V(h,s)$ possesses nonzero variance, while $V(h)$ has zero variance.
>
> > Question 2:
> >
> > How does the method perform over a range of values of $\beta$?
>
> To address your concerns, we conducted ablation experiments on the values of the $\beta$ parameter ($\beta \in \\{1/5, 1/3, 1/2, 2/3, 4/5\\}$). Figure 1 of the rebuttal PDF illustrates that **the choice of $\beta$ exhibits notable stability**. This stability primarily arises from the characteristics of lower-bound-soft-Q-learning. Consequently, we selected $\beta = 1/2$ as the parameter for reporting our results in the paper, as it consistently exhibits superior performance.
>
> Thank you for your thoughtful review and insights into our work, which have significantly improved the quality of our paper. If you find that these concerns have been resolved, we would appreciate it if you would reconsider your ratings of our paper.

---

> ### Comment · Reviewer_VBT3 · 2024-08-08
> **Reviewer Response**
>
> I would like to thank the authors for their detailed responses to my comments. I would also like to congratulate the authors for having produced useful additional results, particularly the ablation on $\beta$ in the short space of time.
>
> My opinion is that overall the paper has merit and the additional analyses have allayed some of my concerns. In light of the fact that parts of the main insights exist albeit within the multi-agent reinforcement literature and that the corresponding methodology may be applied in the degenerate, single-agent case means that the contribution is somewhat constrained. In MARL, there are also techniques to learn how best to interpolate or switch between using a value function with global state inputs and that with local inputs which are worth mentioning e.g. [1]. This is an important consideration as it is likely not a priori known which environments would benefit from global information during training should this information be available.  For this reason, I will keep my score as is.
>
> **Other points**
>
> * I recommend the authors use a different notation for the value functions $V^\pi(h,s)$ and $V^\pi(h)$ and similarly for the action-value functions.
>
> * I think my earlier point about jump-discontinuities can be addressed by showing that $V^\pi_{\rm dual}$ is a smooth function w.r.t. the update parameters of $V^\pi(h,s)$ and $V^\pi(h)$  whenever the latter two functions are differentiable and, showing the corresponding statement for $Q^\pi_{\rm dual}$ .
>
> [1] Mguni, David Henry, et al. "Mansa: Learning fast and slow in multi-agent systems." International Conference on Machine Learning. PMLR, 2023.

---

> ### Author Response · Authors · 2024-08-08
>
> Thank you for your prompt response. We also appreciate your recognition of our supplementary experiment and are glad to have addressed some of your concerns. We will incorporate your suggestions regarding notation and jump discontinuities in our revision of the manuscript. For your remaining questions, we provide point-to-point responses:
>
> > Q1: Regarding the relevance to MARL problems.
>
> We apologize for the misunderstandings that have arisen. The setup presented in our paper differs from the MARL problem formulation.
>
> In MARL, the optimization objective is to maximize the reward under the global states, optimizing each agent accordingly. These methods mostly presuppose that the combination of optimized actions under partial observations leads to optimality under global states. However, this assumption does not always hold, even when $N=1$. The joint-action learner can not be decomposed into independent actors. The policy that maximizes rewards under partial observations does not necessarily align with the policy that maximizes rewards under global states, i.e., $Q(h, a) \neq Q(s, a)$ [DRQN2015Arxiv]. In contrast to MARL, the optimization objective in the POMDP problem discussed in this paper does not rely on this assumption, focusing instead on maximizing rewards under partial observations. Therefore, compared to our DCRL, the applicability of MARL methods in POMDP contexts is limited.
>
> > Q2: Regarding the effectiveness of state information.
>
> We fully understand your concerns. In POMDPs, rewards are directly defined based on state information rather than on observations. Consequently, learning the value function $V(h,s)$ with the reward as the target is typically simpler than learning $V(h)$. This characteristic makes state information effective in most practical applications, a conclusion that numerous previous studies have empirically validated [UAAC2022AAMAS, Suphx2020Arxiv, PerfectDou2022NIPS, Honor-of-Kings2020NIPS]. Additionally, our DCRL leverages the advantages of state information while reducing variance, significantly enhancing training efficiency. Extensive experiments demonstrate that our DCRL consistently outperforms methods that do not incorporate state information across all tested tasks.
>
> Thank you very much for your contribution in clarifying our method. Please feel free to let us know if you have any further concerns.
>
> **Reference**
>
> - [DRQN2015Arxiv] Hausknecht Matthew, et al. Deep recurrent q-learning for partially observable MDPs. Arxiv, 2015.
> - [UAAC2022AAMAS] Andrea Baisero, et al. Unbiased asymmetric reinforcement learning under partial observability. In AAMAS, pages 44–52, 2022.
> - [Suphx2020Arxiv] Junjie Li, et al. Suphx: Mastering mahjong with deep reinforcement learning. Arxiv, 2020.
> - [PerfectDou2022NIPS] Guan Yang, et al. PerfectDou: Dominating Doudizhu with perfect information distillation. In NIPS, pages 34954–34965, 2022.
> - [Honor-of-Kings2020NIPS] Deheng Ye, et al. Towards playing full MOBA games with deep reinforcement learning. In NIPS, pages 621-632, 2020.

---

> ### Comment · Reviewer_VBT3 · 2024-08-12
> **Re:**
>
> Thanks again to the authors for their detailed response.
>
> My comment about the relationship with MARL is that the centralised training with decentralised execution (CT-DE) paradigm closely resembles the central idea being presented, albeit for a distributed setting with a different need to aggregate local information. In particular, the methods that stem from CT-DE degenerate into something similar when considering the case of $N=1$. While it is true that in MARL, one has to consider how to best promote the so-called individual-global-max condition, in the degenerate $N=1$ case this may not be a concern, I believe as the local and joint policy coincide (even if the critic is centalised). Note also that the solution to a POMDP is in general, different to the solution when the agent is endowed with the missing data (which recovers the MDP setup) - this is not specific to the multi-agent setting.
>
> I would like to thank the authors for their response to point 2 - this is a helpful clarification.

---

> ### Author Response · Authors · 2024-08-13
>
> Thank you very much for your feedback! We are glad to have addressed some of your concerns.
>
> **Indeed, there are situations where the local and joint policy coincide, this holds ture when the state encompasses history (rather than just the observation).** The centralized critic is equivalent to using $V(h, s)$, which theoretically ensures that the policy is unbiased and converges to the optimal solution. Since MARL primarily deals with cooperation and competition among multiple agents, partial observability often stems from other agents. When the problem is reduced to $N=1$, most tasks fall into this situation.
>
> **In POMDPs, it is more common for the state not to fully include history**, which is equivalent to using $V(s)$. The state alone does not typically reveal whether the agent has previously collected the necessary information and thus cannot adequately indicate if the current state is favorable or adverse. **The policy is biased and may not converge to the optimal solution.** The UAAC paper has proved it in Theorem 4.2 and includes a toy example to illustrate the concept in Appendix B.2.
>
> We appreciate your constructive feedback and apologize for the lack of clarity in our description. We will provide a more detailed explanation in the revised version of our manuscript.

---

### Official Review · Reviewer_3R7k · 2024-07-13

**Soundness:** 3
**Presentation:** 4
**Contribution:** 4
**Rating:** 7
**Confidence:** 5

**Summary:**

The paper presents Dual Critic Reinforcement Learning (DCRL), a framework designed to handle partial observability in RL. Traditional RL methods often struggle with high variance and instability when relying on full-state information. DCRL addresses this by integrating two critics: an oracle critic with access to complete state information and a standard critic operating within a partially observable context. This dual approach aims to improve efficiency and reduce variance during training, leading to optimized performance in online environments.

**Strengths:**

- This paper is well structured and easy to read. The authors provide thorough analysis and experiment results.
- The paper provides theoretical proof that DCRL reduces learning variance while maintaining unbiasedness. The dual value function is defined as a weighted combination of the oracle and standard value functions, and it is proven to be an unbiased estimate of the true value function with lower variance compared to using the oracle critic alone.
- The effectiveness of DCRL is validated through extensive experiments in the Box2D and Box3D environments. Results show that DCRL significantly outperforms baseline methods, including Recurrent Actor-Critic and Asymmetric Actor-Critic frameworks, across various tasks in the MiniGrid and MiniWorld environments. DCRL achieves faster convergence and higher returns, particularly in complex and high-uncertainty scenarios.

**Weaknesses:**

Refer to Questions.

**Questions:**

- There is a typo in the definition of $R(h,a)$ in line 122.
- Can you explain in more detail how the weighting mechanism between the oracle critic and the standard critic is designed to reduce variance? Are there specific conditions or thresholds for adjusting the weights?
- Is it necessary to ensure the alignment of $V(s)$ and $V(h,s)$ in DCRL? if yes, how to ensure it?

---

> ### Author Rebuttal · Authors · 2024-08-05
>
> Thank you for your encouraging words and constructive feedback. We appreciate your time reviewing our paper and provide point-by-point responses to your comments below.
>
> > Question 1:
> >
> > There is a typo in the definition of $R(h,a)$ in line 122.
>
> We greatly appreciate your pointing out the issues. We have corrected the typos in the revised version.
>
> > Question 2:
> >
> > Can you explain in more detail how the weighting mechanism between the oracle critic and the standard critic is designed to reduce variance? Are there specific conditions or thresholds for adjusting the weights?
>
> To address the high variance issue arising from an over-reliance on state information, we propose a weighting mechanism that leverages the benefits of two critics. As demonstrated in Theorem 2, the no-variance standard critic can effectively mitigate the variance introduced by the oracle critic, thereby reducing overall variance.
>
> **The weight adjustment is governed by the advantage function of the standard critic.** When the advantage function yields a non-positive value, $\beta$ is clipped to $0$. Conversely, when the advantage is positive, $\beta$ activates the standard critic to accelerate training and reduce variance. Theorem 5 establishes that this weighting mechanism is a form of lower-bound-soft-Q-learning. This approach significantly enhances the performance of DCRL while maintaining high stability concerning hyperparameters.
>
> > Question 3:
> >
> > Is it necessary to ensure the alignment of $V(s)$ and $V(h,s)$ in DCRL? if yes, how to ensure it?
>
> Thank you for this question. The answer is no. It is unnecessary to ensure the alignment of $V(s)$ and $V(h,s)$. In the POMDP setting, the policy operates based on partial observations, resulting in a standard policy gradient derived from $V(h)$. Maintaining the policy gradient's unbiasedness is crucial when incorporating state information to enhance training. **This stipulates that the value function must be unbiased concerning $V(h)$ without considering its relationship with $V(s)$.** Our DCRL framework emphasizes two critics: $V(h)$ and $V(h,s)$. It has been established that $V(h,s)$ is unbiased about $V(h)$. Furthermore, Theorem 1 demonstrates that $V_{dual}(h,s)$ in the DCRL framework is also unbiased.
>
> Thank you very much for recognizing our work! Your time and effort are greatly appreciated!

---

> > ### Comment · Reviewer_3R7k · 2024-08-11
> > **Response**
> >
> > Thank you to the authors for answering my questions! I will keep my score.

---

> > > ### Author Response · Authors · 2024-08-11
> > >
> > > Thank you very much for your support! We appreciate the time and effort you've invested in reviewing our manuscript!

---

### Official Review · Reviewer_gCno · 2024-07-14

**Soundness:** 3
**Presentation:** 3
**Contribution:** 2
**Rating:** 5
**Confidence:** 4

**Summary:**

In this work, the authors aim to learn policies in the POMDP setting. They make use of two critics - one that has the privileged state information and one that doesn't and uses only history. Creating a dual value function which is a convex combination of these two value functions. They use this dual value function for policy optimization. They perform experiments in box2d and box3d environments.

**Strengths:**

The authors use the bias-variance tradeoff well. Using standard critic is biased but using only the oracle critic has high variance. The come up with an estimator with low bias as well as low variance and empirically justify this in their experiments.

**Weaknesses:**

(1) Even assuming that you have the state information during training is a strong assumption. You might not always have that. A lot of works do not make that assumption and perform pretty well.

(2) There should have been comparisons to commonly used POMDP works like Dreamer [] and others. But the comparisons were made only with (in some words) different variants of the algorithm.

**Questions:**

(1) Do you think in most real world settings you described in the introduction, you will have access to the state information for training the oracle critic?

**Limitations:**

The authors have discussed their limitation.

---

> ### Author Rebuttal · Authors · 2024-08-05
>
> Thank you very much for your constructive comments and suggestions. We have revised our paper accordingly. Below, we present detailed responses to each point.
>
> > Weakness 1 and Question 1:
> >
> > Even assuming that you have the state information during training is a strong assumption. You might not always have that. A lot of works do not make that assumption and perform pretty well.
> >
> > Do you think in most real world settings you described in the introduction, you will have access to the state information for training the oracle critic?
>
> We fully understand your concerns. To address your concerns, we included a more detailed explanation of the assumptions regarding state information accessibility and provided a representative **real-world scenario (autonomous driving)** of the POMDP problem.
>
> Incorporating state information during the training process is a common assumption adopted by many previous works. In POMDP problems such as MiniGrid and MiniWorld, researchers often use state information to assist agents during training [UAAC2022AAMAS, Believer2023ICML]. In various partially observable games, including **card games (e.g., Mahjong, DouDiZhu) and MOBA games (e.g., Honor of Kings)**, we can readily obtain the state information (opponents' hands or attributes) during training. Consequently, researchers often leverage this information to enhance agent training, resulting in significant performance improvements [Suphx2020Arxiv, PerfectDou2022NIPS, Honor-of-Kings2020NIPS].
>
> In autonomous driving applications, due to the high costs of specific sensors and the challenges of acquiring high-definition maps in some areas, existing solutions mainly rely on end-to-end visual models [Tesla, PhiGent]. During the training phase, autonomous driving companies typically equip a small fleet of training vehicles with expensive sensors and **use the collected high-definition maps and other information as state information to facilitate the training of these end-to-end visual models**. This approach aids in constructing 3D environments, thereby enhancing the performance of autonomous driving systems [BEV2023TPAMI].
>
> > Weakness 2:
> >
> > There should have been comparisons to commonly used POMDP works like Dreamer [] and others. But the comparisons were made only with (in some words) different variants of the algorithm.
>
> Thank you for this valuable suggestion. We have followed your advice and considered the commonly used POMDP baseline DreamerV3 [DreamerV32023Arxiv].  We compared the performance of DCRL, DreamerV3, and additional baselines across eight tasks in the MiniGrid environment. Figure 3 of the rebuttal PDF illustrates the average scores from five training seeds over $1e5$ steps for each method. **The results indicate that DCRL consistently outperforms DreamerV3 across all eight tasks.** Importantly, DCRL requires fewer computational resources and less training time than DreamerV3.
>
> Thank you for your thought-provoking and discussion-worthy suggestions, which have significantly improved the quality of our paper. If you find that these concerns have been resolved, we would appreciate it if you would consider reflecting this in your rating of our paper.
>
> **References**
>
> - [UAAC2022AAMAS] Andrea Baisero, et al. Unbiased asymmetric reinforcement learning under partial observability. In AAMAS, pages 44–52, 2022.
> - [Believer2023ICML] Andrew Wang, et al. Learning belief representations for partially observable deep RL. In ICML, pages 35970–35988, 2023.
> - [Suphx2020Arxiv] Junjie Li, et al. Suphx: Mastering mahjong with deep reinforcement learning. Arxiv, 2020.
> - [PerfectDou2022NIPS] Guan Yang, et al. PerfectDou: Dominating Doudizhu with perfect information distillation. In NIPS, pages 34954–34965, 2022.
> - [Honor-of-Kings2020NIPS] Deheng Ye, et al. Towards playing full MOBA games with deep reinforcement learning. In NIPS, pages 621-632, 2020.
> - [Tesla] Tesla AI Day. [Online]. Available: https://www.youtube.com/watch?v=j0z4FweCy4M
> - [PhiGent] PhiGent: Technical Roadmap. [Online]. Available: https://43.132.128.84/coreTechnology
> - [BEV2023TPAMI] Hongyang Li, et al. Delving into the devils of bird's-eye-view perception: A review, evaluation and recipe. IEEE TPAMI, 2023.
> - [DreamerV32023Arxiv] Hafner Danijar, et al. Mastering diverse domains through world models. Arxiv, 2023.

---

> > ### Comment · Reviewer_gCno · 2024-08-11
> >
> > Thank you for your response and additional experiments. I am increasing my score based on the clarifications. I agree that there might be environments where obtaining state-information is possible and you can use that state observation during training but that would severely limit your work in terms of applicability to any POMDP setting.

---

> > > ### Author Response · Authors · 2024-08-12
> > >
> > > We sincerely appreciate the time and effort you have invested to the discussions! The above inspiring discussion has greatly enhanced the quality of our paper. The scope of our study encompasses situations where state information is accessible during training, which is an essential topic in POMDPs with many applications like autonomous driving. For problem settings without access to state information, we can leverage a pre-trained teacher network to provide labels, thereby facilitating training without compromising deployment. We will incorporate this suggestion and provide a more comprehensive discussion in the revised manuscript.
> > >
> > > --Best wishes from all the authors!

---

### Official Review · Reviewer_Hmpg · 2024-07-15

**Soundness:** 4
**Presentation:** 3
**Contribution:** 2
**Rating:** 6
**Confidence:** 4

**Summary:**

This paper proposes a dual critic architecture for learning POMDPs. One critic is the standard critic that uses the history information while the other critic is the unbiased asymmetric critic that uses the history and state information. The authors prove that using both critics reduces variance and propose a weighting mechanism between the two critics based on the advantage. Empirical results show that DCRL provides a substantial benefit over the asymmetric critic or unbiased asymmetric critic.

**Strengths:**

- The authors motivate the problem clearly and explain why a dual critic approach would be useful
- The authors prove that variance is reduced
- Experimental results show significant improvements over baselines on minigrid and miniworld.

**Weaknesses:**

The primary weakness of this paper is novelty. While I can appreciate that the dual critic architecture reduces variance, the proofs seems fairly obvious. In my opinion, the main novel contribution of this paper then relies on the weighting mechanism and the empirical results showing that DCRL can beat AC, UAAC, and also recurrent A2C. The novelty therefore seems marginal.

Additionally, asymmetric actor critic (AC) was proposed to utilize additional state information to learn a policy more efficiently over the standard recurrent A2C where the critic only uses history information. UAAC then corrects the bias in AC by incorporating both the state and the history. However this paper proposes going back to the standard recurrent A2C method for one of the critics to achieve unbiasedness. It would have been interesting to compare two dual critic architectures: 1) V(h) and V(h,s) and 2) V(s) and V(h,s). Although the second approach would be biased, I would assume that using V(s) would lead to more efficiency gains. This seems to be an obvious comparison that was not considered.

**Questions:**

See weaknesses

**Limitations:**

The authors clearly discuss the limitations of their work.

---

> ### Author Rebuttal · Authors · 2024-08-05
>
> We appreciate your constructive comments, which have significantly enhanced the quality of our manuscript. Below, we provide a point-by-point response to your feedback.
>
> > Weakness 1:
> >
> > The primary weakness of this paper is novelty. While I can appreciate that the dual critic architecture reduces variance, the proofs seems fairly obvious. In my opinion, the main novel contribution of this paper then relies on the weighting mechanism and the empirical results showing that DCRL can beat AC, UAAC, and also recurrent A2C. The novelty therefore seems marginal.
>
> Thank you for your valuable feedback. In this study, we address the critical issue of high variance resulting from an over-reliance on state information, **a topic that has been largely overlooked but is highly relevant to the POMDP community**. To mitigate this issue, we propose a DCRL framework that harnesses the strengths of two critics. We provide theoretical evidence demonstrating that DCRL mitigates the learning variance while maintaining unbiasedness. Moreover, our DCRL incorporates **a dynamic weighting mechanism**, which can be interpreted as a form of **lower-bound-soft-Q-learning**, distinguishing it from conventional linear weighting methods. This mechanism substantially improves DCRL performance while maintaining **high stability concerning hyperparameters**.
>
> We conducted additional experiments to validate our claims: 1) We implemented a simplified weighting method that uses fixed ratios to weight the two critics (referred to as the DCRL No Clip Version). 2) We performed ablation experiments on the values of the $\beta$ parameter ($\beta \in \\{1/5, 1/3, 1/2, 2/3, 4/5\\}$). Figure 1 of the rebuttal PDF illustrates that the DCRL No Clip version shows performance improvements in only some environments compared to UAAC. In contrast, DCRL consistently enhances performance across all tested environments, **demonstrating that the dynamic weighting mechanism of DCRL is crucial. Additionally, the choice of $\beta$ exhibits notable stability**. We selected $\beta = 1/2$ as the parameter for reporting our results in the paper, as it consistently exhibits superior performance.
>
> > Weakness 2:
> >
> > Additionally, asymmetric actor critic (AC) was proposed to utilize additional state information to learn a policy more efficiently over the standard recurrent A2C where the critic only uses history information. UAAC then corrects the bias in AC by incorporating both the state and the history. However this paper proposes going back to the standard recurrent A2C method for one of the critics to achieve unbiasedness. It would have been interesting to compare two dual critic architectures: 1) V(h) and V(h,s) and 2) V(s) and V(h,s). Although the second approach would be biased, I would assume that using V(s) would lead to more efficiency gains. This seems to be an obvious comparison that was not considered.
>
> We greatly appreciate your suggestion. Following your suggestion, we conducted **ablation experiments comparing two different dual architectures**: 1) $V(h)$ and $V(h,s)$ (our DCRL), and 2) $V(s)$ and $V(h,s)$ (referred to as the DCRL State Version). Figure 2 of the rebuttal PDF illustrates that the DCRL State Version demonstrates commendable performance among the various DCRL variants. However, the biased characteristics of the DCRL State Version contribute to instability in its effectiveness. **Notably, our DCRL achieves superior training performance in three out of four environments.** These findings further highlight the significance of our DCRL framework.
>
> We appreciate your constructive feedback. These questions are very inspiring and discussion-worthy. The quality of the manuscript has been greatly enhanced by incorporating these and other reviewers' valuable suggestions. If you find that these concerns have been resolved, we would appreciate it if you would consider reflecting this in your rating of our paper.

---

> > ### Comment · Reviewer_Hmpg · 2024-08-08
> >
> > I thank the authors for their detailed and focused responses. I am honestly impressed that they were able to perform all of the new experiments and the new results strengthen the paper greatly. I've raised my score.
> >
> > I'm now more convinced that DCRL is a good approach and outperforms other possible reasonable methods. In particular, the fixed weighting $\beta$ experiments show that dynamic reweighting is crucial, increasing this paper's contribution.
> >
> > One minor point in the rebuttal is that I'm not convinced of the argument that the mechanism resembles lower-bound-soft-Q-learning since soft Q learning fundamentally optimizes a different objective (max entropy RL or soft Bellman equation), while here the critic is still optimized for the standard Bellman equation, but again this is minor.

---

> > > ### Author Response · Authors · 2024-08-09
> > >
> > > We sincerely appreciate the time and effort you have dedicated to the discussions! The above inspiring discussion has greatly improved the quality of our paper. Thank you for recognizing our efforts. We want to assure you that we are committed to incorporating these discussions into the final version.
> > >
> > > Regarding the lower-bound soft-Q-learning, we employ an approximation in which the entropy parameter approaches zero to meet the condition. We will incorporate this suggestion and provide a more comprehensive discussion in the revised manuscript.
> > >
> > > --Best wishes from all the authors!

---

### Author Rebuttal · Authors · 2024-08-05

We appreciate all the reviewers for their insightful and constructive feedback. In response to these helpful comments, we conducted supplementary experiments in the MiniGrid environment (see the attached PDF):

1. **Figure 1**: Ablation studies on different values of $\beta \in \\{1/5, 1/3, 1/2, 2/3, 4/5\\}$ and a simple weighting method (referred to as DCRL No Clip Version) with fixed ratios to weight the two critics.
2. **Figure 2**: Comparison of two different dual architectures: 1) $V(h)$ and $V(h,s)$ (our DCRL), and 2) $V(s)$ and $V(h,s)$ (referred to as DCRL State Version).
3. **Figure 3**: Comparison with the commonly used POMDP baseline, DreamerV3.

These experiments further demonstrate the following:

1. The dynamic weighting mechanism of DCRL is crucial, and the choice of $\beta$ exhibits notable stability.
2. Our DCRL framework outperforms the DCRL State Version due to its unbiased characteristic, which contrasts with the biased nature of the DCRL State Version.
3. DCRL consistently surpasses DreamerV3 across all tasks, confirming its superior performance.

We hope these additional experimental results provide further evidence of the effectiveness of our DCRL framework.

Moreover, we made several revisions and improvements to our paper:

1. We included a more detailed explanation of the assumptions regarding state information accessibility in the introduction section.
2. We addressed the degenerate case of CTDE when $N=1$ in the related work section.
3. We added additional toy examples in the appendix to illustrate when DCRL is effective and when it may fail.

Furthermore, we addressed each reviewer's questions or concerns with detailed point-by-point responses. As a result, the quality of our paper has markedly improved.

We are profoundly grateful for the reviewers' contributions to our paper and sincerely hope that our revisions have adequately addressed your concerns. We remain open to further feedback and are committed to implementing additional improvements if necessary.

---

### Decision · Program_Chairs · 2024-09-25

**Decision:**

Accept (poster)

**Comment:**

The paper proposes a novel partially observable RL method for settings when the state is available during training. The approach combines a history-state critic (which has been shown to perform well) with a history critic using a dynamic weighting scheme to reduce variance and improve performance. The paper theoretically shows that variance can be reduced and the method remains unbiased. The approach works well in a number of domains using A2C and PPO as base methods.

Asymmetric critics that incorporate privileged information in training are becoming common in partially observable RL (RL for POMDPs). While the current state-of-the-art methods typically use history-state, the variance can be high. The proposed approach is relatively simple but well-motivated, simple to implement, and performs well.

The author response was helpful for better clarifying the method and showing the benefits of the method, including the use of dynamic weighting and against other baselines. These results and discussions should be included in the final paper.